# TempFlex: Advancing MLLMs with Temporal Perception and Natively Scalable Resolution Encoding

**Zhanyu Wang**                                     *zhanyu.wang@bytedance.com*
*ByteDance*

**Chen Tang**[*]                                     *tangchen.222@bytedance.com*
*ByteDance*

**Haoyu He**                                         *haoyu.he@bytedance.com*
*ByteDance*

**Kuan Feng**                                        *fengkuan.123@bytedance.com*
*ByteDance*

**Chao Wang**                                        *wangchao.hhhhh@bytedance.com*
*ByteDance*

**Bingni Zhang**                                     *bingni.zhang@bytedance.com*
*ByteDance*

**Xiaolei Xu**                                       *xuxiaolei@bytedance.com*
*ByteDance*

**Shen Wang**[†]                                     *wangshen.summer@bytedance.com*
*ByteDance*

**Luping Zhou**[†]                                   *luping.zhou@sydney.edu.au*
*The University of Sydney*

**Reviewed on OpenReview:** *https://openreview.net/forum?id=ietYdtRB3h*

## Abstract

Multimodal large language models (MLLMs) have made significant progress across vision-language tasks, yet many designs still suffer from two core limitations. (i) Excessive visual tokens and broken global context: Tiled Patch Encoding fragments high-resolution images, leading to token overload and disrupting global attention modeling. (ii) Lack of temporal reasoning: Most models process video as independent frames using static image encoders, failing to capture temporal dynamics. We present TempFlex-VL, a token-efficient and temporally aware MLLM that addresses both issues through lightweight architectural enhancements. First, we introduce a resolution-agnostic visual encoder that directly processes full images without tiling, preserving global context while substantially reducing visual tokens. Second, we propose Temporal Fiber Fusion (TFF), a plug-and-play module with three complementary pathways: (1) a dynamic local-convolution branch for fine-grained motion, (2) a gated memory accumulator for long-term dependencies, and (3) a periodic encoder for modeling cyclic patterns. These signals are softly fused, enabling the model to adapt to diverse temporal structures without overfitting. To support large-scale video-language pretraining, we curate TempFlex-2M, a high-quality synthetic video–text corpus generated in a single stage via GPT-4o with direct visual prompting. We instantiate TempFlex-VL using two

---

[*]Project Lead.
[†]Corresponding Authors.

different language backbones, Gemma3-4B and Qwen3-4B, demonstrating the generality of our design across architectures. Both variants achieve state-of-the-art or competitive results on a wide range of image and video benchmarks while markedly improving token efficiency. Code is publicly available at: https://github.com/wang-zhanyu/TempFlex.

# 1 Introduction

Multimodal large language models (MLLMs) (Zohar et al., 2024; Li et al., 2023a; Tong et al., 2024; Lu et al., 2024a; Wu et al., 2024b; Chen et al., 2025; Gu et al., 2024; Liu et al., 2023; Zhu et al., 2023; Liu et al., 2024a; Li et al., 2024a; Ye et al., 2024; Team, 2024; Chen et al., 2024e; Zhu et al., 2025; Wang et al., 2024a; Bai et al., 2025; Liu et al., 2024b; Abdin et al., 2024a;b; Deitke et al., 2024; Liu et al., 2024e; Lu et al., 2024b; Lin et al., 2024; Cho et al., 2025; Wang et al., 2024b; Yao et al., 2024; Liu et al., 2024f; Zhang et al., 2024a; Chen et al., 2024d;c; Lin et al., 2023a) have emerged as a powerful foundation for general-purpose AI systems, enabling unified understanding and reasoning across text, images, and videos. These models demonstrate impressive performance in tasks such as visual question answering, image and video captioning, visual grounding, and open-ended dialog grounded in visual context. As research shifts toward longer, higher-resolution, and temporally complex visual inputs, improving the scalability and efficiency of MLLMs becomes increasingly critical.

Most existing MLLMs (Wu et al., 2024b; Chen et al., 2024e; Zhu et al., 2025; Liu et al., 2024a; Li et al., 2024a; Wang et al., 2025; Cho et al., 2025) follow a standard architecture popularized by the LLaVA (Liu et al., 2023) framework. In this paradigm, visual inputs are first processed by a vision encoder (e.g., CLIP (Radford et al., 2021), SigLIP (Zhai et al., 2023; Tschannen et al., 2025)) to extract per-frame image embeddings. These embeddings are then projected into the language model's token space via a vision-language connector module, after which the LLM performs cross-modal reasoning over the joint sequence of visual and textual tokens. Early approaches such as BLIP-2 (Li et al., 2023a) and LLaVA (Liu et al., 2023) adopted a fixed-resolution encoding strategy, where all input images are resized to a pre-defined resolution (e.g., $384\times384$) prior to feature extraction. This resizing distorts the aspect ratio and potentially damages the spatial integrity of high-resolution images, leading to information loss. To alleviate this issue, recent works like LLaVA-Next (Liu et al., 2024a) introduced a tiled patch encoding mechanism: each image is decomposed into multiple overlapping tiles (e.g., $384\times384$), each tile is independently encoded, and the resulting embeddings are concatenated along with the embedding of the full image. While this improves high-resolution coverage, it introduces two major limitations: 1) the lack of global spatial attention due to independent encoding of tiles, and 2) substantial visual token redundancy, which significantly increases computational overhead. Moreover, while there have been emerging efforts to extend MLLMs to video-level tasks (Wang et al., 2025; Cho et al., 2025), most models still rely on static image encoders applied frame-by-frame, without explicit temporal modeling. This neglect of temporal dynamics—such as motion patterns, causal relations, and temporal dependencies—hinders their ability to reason over videos effectively. As a result, current MLLMs struggle to jointly understand both high-resolution spatial structures and temporal semantics in visual content.

In this work, we present TempFlex-VL, a token-efficient and temporally-aware MLLM designed to address both spatial and temporal limitations in existing pipelines. As shown in Figure 1, TempFlex-VL delivers strong results across a variety of challenging image and video understanding tasks. To improve image encoding, we propose a naive any-resolution strategy based on SigLIP2 (Tschannen et al., 2025). Rather than tiling high-resolution images into fixed-size crops, we modify the encoder to directly process images of arbitrary resolution while preserving full spatial structure. Compared to tiled-patch encoding, our method—integrated with patch merger—significantly reduces the number of visual tokens (by at least 72%) while maintaining spatial fidelity and enhancing downstream task performance, as demonstrated in the results shown in Figure 3.

For temporal modeling, we introduce Temporal Fiber Fusion, a lightweight, training-friendly module that augments frame-wise image features with explicit temporal structure. TFF operates on the temporal axis of per-patch features (i.e., shape $[B, F, P, D]$) and can be appended after a image encoder in a fully plug-and-play manner. It comprises three parallel temporal pathways ("fibers") that capture complementary motion patterns: (i) a *Local fiber* applies dynamic $1\times3$ convolutions using each patch's mean embedding to capture fine-grained motion over adjacent frames; (ii) a *Memory fiber* maintains a gated accumulator whose

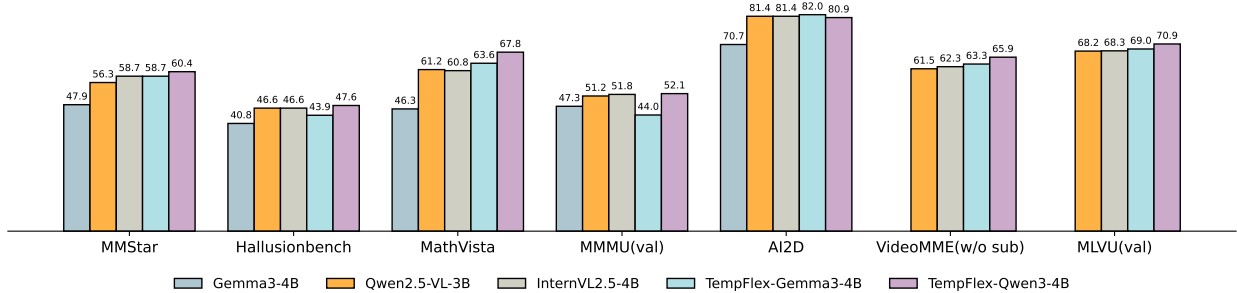

Figure 1: Performance Comparison of TempFlex-VL with the previous advanced image/video MLLMs on various representative benchmarks.

update rate is controlled by learned gates $\alpha = \sigma(Wx)$, enabling integration of long-term dependencies and static context; and (iii) a *Periodic fiber* encodes frame positions using complex exponentials across multiple frequencies to detect recurring patterns such as flashes or oscillations. A learned fusion layer dynamically weights these pathways per video sequence, allowing the model to adapt to diverse motion characteristics while reducing the risk of overfitting.

We push the limit of large-scale video-language pretraining by introducing TempFlex-2M, a synthetic multimodal video dataset containing 2 million samples. Previous pipelines for creating synthetic data for video instruction tuning generally follow a multi-stage approach (Zhang et al., 2024b;c): they first generate frame-level image captions, and then prompt large language models to produce question-answer pairs based on these captions. However, this process inevitably introduces an information bottleneck during caption generation. In contrast, our TempFlex-2M is constructed in a single stage by prompting GPT-4o with visual input directly, producing temporally aligned and semantically rich narrations.

**Our main contributions are summarized as follows:**

(1) We propose an any-resolution visual encoding strategy that processes high-resolution images directly without tiling, preserving global context and reducing visual tokens by at least 72% compared to tiled patch encoding, while achieving favorable accuracy-efficiency trade-off.

(2) We design a plug-and-play Temporal Fiber Fusion module to enhance temporal modeling across video frames, which quickly adapts to pre-trained visual encoders.

(3) We construct a large-scale and high-quality synthetic video dataset TempFlex-2M tailored for multimodal video-language pretraining, which brings consistent improvements across multiple video benchmarks.

(4) We demonstrate the generality and efficiency of TempFlex-VL by instantiating it with Gemma3-4B and Qwen3-4B backbones, achieving competitive or state-of-the-art performance across diverse image and video understanding benchmarks (Figure 1), with all code, models, and data to be released publicly.

## 2 Related Works

**Vision-language models.** Leveraging the capabilities of large language models (LLMs), a growing body of Multimodal large language models (MLLMs) has been introduced for image understanding (Li et al., 2023a; Gu et al., 2024; Liu et al., 2023; 2024a; Lin et al., 2024; Gao et al., 2024b; Deitke et al., 2024; Abdin et al., 2024a; Jeddi et al., 2025), video understanding (Zohar et al., 2024; Lin et al., 2023b; Shu et al., 2024; Li et al., 2023b; Wang et al., 2025; Chen et al., 2024d; Zhang et al., 2024a; Wang et al., 2024b; Jiang et al., 2025), and unified image-video comprehension (Wu et al., 2024b; Chen et al., 2024e; Zhu et al., 2025; Wang et al., 2024a; Bai et al., 2025; Chen et al., 2025; Li et al., 2024a; Cho et al., 2025; Gao et al., 2024a). These models incorporate advances such as dynamic high-resolution inputs (Li et al., 2024a), adaptive token compression (Jiang et al., 2025; Jeddi et al., 2025), and multimodal positional embeddings (Wang et al., 2024a; Bai et al., 2025), enabling more fine-grained image and video understanding. In the same spirit, we

introduce TempFlex-VL, a unified vision-language model designed for both image and video understanding, featuring a token-efficient any-resolution encoding strategy and a plug-and-play temporal fusion module to achieve state-of-the-art results across a broad range of MLLMs tasks.

**Multimodal synthetic dataset.** Training data is critical for MLLM development, with synthetic data emerging as a key ingredient for instruction tuning (Abdin et al., 2024b). While many methods exist for image-based instruction datasets (Zhao et al., 2024; Yang et al., 2025b; Liu et al., 2023; Deitke et al., 2024; Chen et al., 2024a; Su et al., 2023), synthetic video-based data remains scarce and less diverse (Zhang et al., 2024c;b). Recent approaches like LLaVA-Video-178K (Zhang et al., 2024c) follow a two-stage pipeline: first generating detailed captions using GPT-4o (Hurst et al., 2024), followed by constructing VQA pairs based solely on the generated captions. However, this may overlook fine-grained visual or temporal details, leading to semantically biased supervision. To overcome this, we introduce TempFlex-2M, a large-scale synthetic dataset where GPT-4o (Hurst et al., 2024) directly processes raw visual content and auxiliary text (e.g., OCR, ASR, titles) to generate VQA data in a single stage, enhancing visual grounding and temporal richness.

## 3 Method

### 3.1 Network Architecture

As illustrated in Figure 2, our architecture consists of four main components: a visual encoder, a temporal Fiber Fusion module for temporal modeling, a Patch Merger module for visual token compression, and a large language model.

**Visual Encoder.** We adopt the pretrained SigLIP2-patch16-naflex as our visual encoder. Unlike patch-based approaches that require fixed input sizes or tiling (e.g., 384×384 crops), SigLIP2-naflex natively supports arbitrary input resolutions by design. This is enabled by its flexible positional encoding mechanism and the ViT-style patch embedding. To better accommodate high-resolution images, we extend the original positional embedding length from 256 to 2048 using bilinear interpolation:

$$p'_j = \text{Interp}(p, j), \quad j = 1, \ldots, 2048,$$

where $p$ is the pretrained positional embedding. This extension enables efficient encoding of large images while preserving spatial context. Details of the visual encoding process are provided in Appendix A.1.

**Temporal Fiber Fusion Module.** As SigLIP2 is trained for still image understanding, it lacks temporal modeling capabilities necessary for video tasks. To address this, we introduce a lightweight Temporal Fiber Fusion module to enhance temporal reasoning across frames. TFF captures short-term motion, long-range dependencies, and periodic patterns through specialized temporal pathways. Details are provided in Section 3.2.

**PatchMerger.** To reduce the number of visual tokens and improve token efficiency, we apply a pixel unshuffle operation on the image feature maps. This reduces spatial dimensions while preserving information in the channel dimension. The resulting compact tokens are then projected via a two-layer MLP to match the input space of the language model:

$$z' = \text{MLP}(\text{Unshuffle}(z)),$$

where $z$ are the patch-level features from SigLIP2.

**Large Language Model.** We use the pretrained Gemma3 (Team et al., 2025) and Qwen3 (Yang et al., 2025a) model as our language backbone. Text is tokenized and embedded, then concatenated with the processed visual tokens before being passed to the LLM. The model is trained using a standard cross-entropy loss for next-token prediction.

### 3.2 Temporal Fiber Fusion

We propose `Temporal Fiber Fusion`, a lightweight yet expressive module that injects temporal reasoning into the frame–patch representations produced by the vision encoder. Let $\mathbf{x} \in \mathbb{R}^{B \times F \times P \times D}$ denote the batch

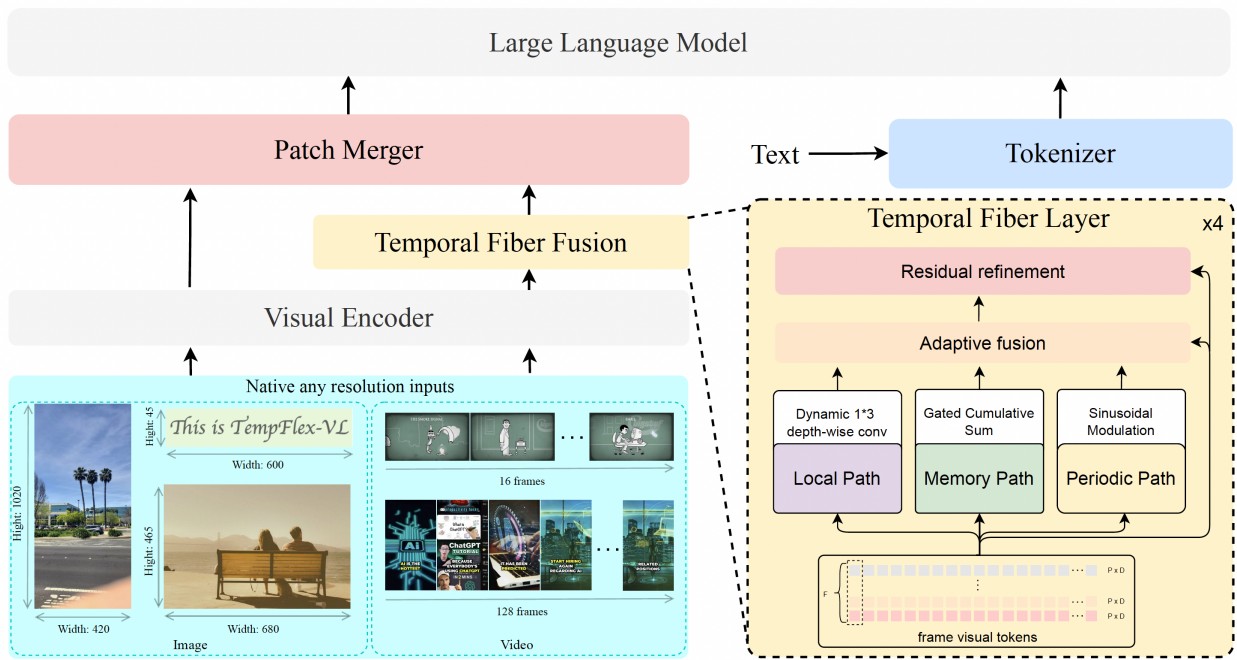

Figure 2: Overview of TempFlex-VL framework. TempFlex natively accommodates image and video inputs of arbitrary resolutions. Its Temporal Fiber Fusion module reinforces temporal modeling, while the Patch Merger effectively reduces the dimensionality of visual tokens.

of visual tokens, where $B$ is the batch size, $F$ the number of frames, $P$ the number of spatial patches per frame, and $D$ the feature dimension.

**Intuition.** For a fixed patch index $p$, the slice $\mathbf{x}_p \triangleq (\mathbf{x}_{1,p}, \ldots, \mathbf{x}_{F,p}) \in \mathbb{R}^{F \times D}$ is a *temporal fiber* describing how that spatial location evolves over time. TFF treats each fiber as a one-dimensional signal and derives three orthogonal temporal viewpoints: *local*, *memory*, and *periodic*. These viewpoints are fused with data-dependent weights and finally broadcast across patches to share frame-level context.

### 3.2.1 Module Formulation

For every fiber $p$ we derive three complementary temporal views and fuse them adaptively.

**Local Path.** A two-layer MLP $\phi_{\text{loc}} \colon \mathbb{R}^D \to \mathbb{R}^{3D}$ maps the fibre mean $\bar{\mathbf{x}}_p = \frac{1}{F} \sum_f \mathbf{x}_{f,p}$ to a depth-wise kernel $\mathbf{k}_p = \big[\mathbf{k}_{p,-1}, \mathbf{k}_{p,0}, \mathbf{k}_{p,1}\big] \in \mathbb{R}^{3 \times D}$. The local output is the circular convolution

$$\mathbf{L}_{f,p} = \sum_{j=-1}^{1} \mathbf{k}_{p,j} \odot \mathbf{x}_{(f+j) \bmod F, \, p}. \tag{1}$$

where $\odot$ denotes element-wise, channel-wise multiplication.

**Memory Path.** A gated accumulator captures slow trends. With trainable matrix $\mathbf{W}_\alpha \in \mathbb{R}^{D \times D}$ we set $\alpha_{f,p} = \sigma(\mathbf{W}_\alpha \mathbf{x}_{f,p}) \in (0,1)^D$ where $\sigma(\cdot)$ is the element-wise logistic sigmoid and update

$$\mathbf{m}_{f,p} = \alpha_{f,p} \odot \mathbf{m}_{f-1,p} + (1 - \alpha_{f,p}) \odot \mathbf{x}_{f,p}, \qquad \mathbf{M}_{f,p} = \mathbf{m}_{f,p}, \tag{2}$$

initialised with $\mathbf{m}_{-1,p} = 0$.

**Periodic Path.** A small dictionary of integer frequencies $\Omega = \{\omega_1, \ldots, \omega_K\}$ ($K{=}5$ in practice) detects rhythmic motion:

$$\mathbf{P}_{f,p} = \sum_{\omega \in \Omega} \cos\!\big(2\pi\omega f/F\big)\, \mathbf{W}_\omega \mathbf{x}_{f,p}, \tag{3}$$

where $\mathbf{W}_\omega \in \mathbb{R}^{D \times D}$ are learned projections.

**Adaptive fusion and contextual broadcast.** Fibre-wise weights $\boldsymbol{\beta}_p = \mathrm{softmax}\big(\mathbf{W}_\beta \bar{\mathbf{x}}_p\big) \in \mathbb{R}^3$ combine the three paths:

$$\mathbf{y}_{f,p} = \beta_p^{(1)}\mathbf{L}_{f,p} + \beta_p^{(2)}\mathbf{M}_{f,p} + \beta_p^{(3)}\mathbf{P}_{f,p}. \tag{4}$$

Restoring the original layout yields $\mathbf{Y} \in \mathbb{R}^{B \times F \times P \times D}$. For each frame $f$ we compute the global context $\mathbf{c}_f = \frac{1}{P}\sum_p \mathbf{Y}_{f,p}$ and broadcast it with a learnable scale $\boldsymbol{\gamma} \in \mathbb{R}^D$: $\mathbf{Y}_{f,p} \leftarrow \mathbf{Y}_{f,p} + \boldsymbol{\gamma} \odot \mathbf{c}_f$.

**Residual refinement.** A gated feed-forward network $\mathrm{FFN}(\mathbf{z}) = \mathrm{GLU}\big(\mathbf{W}_2\, \mathrm{GELU}(\mathbf{W}_1\mathbf{z})\big)$ further mixes channels. The final output of one TFF layer is

$$\mathbf{Z}_{f,p} = \mathbf{X}_{f,p} + \mathrm{LayerNorm}\Big(\mathbf{Y}_{f,p} + \mathrm{FFN}(\mathbf{Y}_{f,p})\Big). \tag{5}$$

### 3.3 Vision-Prompted Dataset Construction

To facilitate the training of TempFlex-VL, we construct a high-quality video–language instruction dataset tailored for diverse temporal and visual reasoning tasks. Existing datasets such as LLaVA-Hound (Zhang et al., 2024b), ShareGPT4Video (Chen et al., 2024c), and LLaVA-Video-178k (Zhang et al., 2024c) typically follow a two–stage generation pipeline: an LLM first generates a dense caption from video content, and subsequently produces VQA pairs conditioned solely on that caption. This *Caption → VQA* strategy suffers from information bottlenecks, as unmentioned visual cues—such as fine-grained motions, or non-salient interactions—are irrevocably discarded, leading to generic or context-insensitive supervision.

In contrast, we propose a single–stage *Video+Text → Caption+VQA* pipeline. Given raw video frames along with auxiliary metadata (e.g., titles, tags, and ASR transcripts where available), we prompt GPT-4o to jointly produce (i) a concise clip description and (ii) a diverse set of question–answer pairs. These QA pairs are designed to span twelve complementary reasoning skills: Object Description, Causal Reasoning, Temporal Reasoning, Scene Description, Human Activity, Appearance Description, Spatial Awareness, Sequence Ordering, Attribute Evolution, Count and Quantity, Narrative Understanding, Fine-grain Action Recognition, and Motion Analysis. This direct vision–prompting strategy preserves fine-grained information and yields high-utility, multimodal instruction data.

Our resulting dataset, `TempFlex-2M`, comprises 210K curated and de-duplicated videos sourced from FineVideo (Farré et al., 2024), OpenVid-1M (Nan et al., 2024), Vatex (Wang et al., 2019), and Virpt (Yang et al., 2024). The dataset contains 1.77 million vision-grounded instruction pairs, covering approximately 1,500 hours of footage across domains such as everyday life, sports, how-to videos, documentaries, and short films. A comparative summary is presented in Table 1, and detailed category-wise distributions and the prompts used for data generation are provided in the Appendix A.2.

Table 1: Comparison of TempFlex-2M and other video-language datasets.

| Datasets | Pipeline | Video | Total Video Length | Average FPS | Caption | VQA |
|---|---|---|---|---|---|---|
| LLaVA-Hound (Zhang et al., 2024b) | Video → Caption → VQA | 900K | 3K hr | 0.008 | 900K | 900K |
| ShareGPTVideo (Chen et al., 2024c) | Video → Caption → VQA | 40K | 0.2K hr | 0.15 | 40K | 0 |
| LLaVA-Video-178k (Zhang et al., 2024c) | Video → Caption → VQA | 178K | 2K hr | 1 | 178K | 960K |
| TempFlex-2M | Video + Text → Caption + VQA | 210K | 1.5K hr | 1 | 210K | 1.77M |

Table 2: Comparison with state-of-the-art methods on image benchmarks. The best results are highlighted in **bold**, and the second-best results are underlined. Most of the comparison results in the table are sourced from OpenCompass (Contributors, 2023); for entries not covered by OpenCompass, as well as all our TempFlex results, we performed a consistent evaluation using VLMEvalKit (Duan et al., 2024).

| Models | OCR-related Understanding | | | College-level | Math |
|---|---|---|---|---|---|
| | $AI2D_{test}$ | $DocVQA_{val}$ | TextVQA | $MMMU_{val}$ | $MathVista_{mini}$ |
| Paligemma-3B (Beyer et al., 2024) | 68.3 | 73.9 | 70.1 | 34.9 | 28.5 |
| Gemma3-4B (Team et al., 2025) | 70.7 | 62.3 | 60.3 | 47.3 | 46.3 |
| Phi3.5-vision-4B (Abdin et al., 2024a) | 77.8 | 70.3 | 65.2 | 44.6 | 43.3 |
| Phi4-vision-5.6B (Abdin et al., 2024b) | **83.0** | 92.6 | 76.9 | **56.0** | 65.8 |
| Qwen2-VL-2B (Wang et al., 2024a) | 74.7 | 89.2 | **79.9** | 42.2 | 48.0 |
| Qwen2.5-VL-3B (Bai et al., 2025) | 81.4 | **93.0** | 79.3 | 51.2 | 61.2 |
| Mini-InternVL-Chat-4B (Gao et al., 2024b) | 77.0 | 86.7 | 73.0 | 45.1 | 55.0 |
| InternVL2-4B (Team, 2024) | 79.1 | 88.1 | 74.7 | 48.3 | 58.5 |
| InternVL2.5-4B (Chen et al., 2024e) | 81.4 | 91.1 | 78.8 | 51.8 | 60.8 |
| TempFlex-Gemma3-4B (Ours) | 82.0 | 90.1 | 77.8 | 44.0 | 63.6 |
| TempFlex-Qwen3-4B (Ours) | 80.9 | 91.0 | 79.1 | 52.1 | **67.8** |

| Models | General Visual Question Answering | | | | Overall |
|---|---|---|---|---|---|
| | MMStar | $MMBench\text{-}EN_{dev}$ | Realworldqa | HallBench | |
| Paligemma-3B (Beyer et al., 2024) | 48.3 | 65.6 | 55.2 | 32.2 | 53.0 |
| Gemma3-4B (Team et al., 2025) | 47.9 | 66.4 | 55.6 | 40.8 | 55.3 |
| Phi3.5-vision-4B (Abdin et al., 2024a) | 47.5 | 67.4 | 53.6 | 40.5 | 56.7 |
| Phi4-vision-5.6B (Abdin et al., 2024b) | 58.9 | 77.2 | 64.1 | 40.5 | 68.3 |
| Qwen2-VL-2B (Wang et al., 2024a) | 47.5 | 72.2 | 60.7 | 42.4 | 61.9 |
| Qwen2.5-VL-3B (Bai et al., 2025) | 56.3 | 76.8 | 65.5 | 46.6 | 67.9 |
| Mini-InternVL-Chat-4B (Gao et al., 2024b) | 53.1 | 69.7 | 60.8 | 43.0 | 62.6 |
| InternVL2-4B (Team, 2024) | 53.9 | 73.6 | 60.5 | 42.4 | 64.3 |
| InternVL2.5-4B (Chen et al., 2024e) | 58.7 | 78.2 | 64.6 | 46.6 | 68.0 |
| TempFlex-Gemma3-4B (Ours) | 58.7 | **79.2** | 64.6 | 43.9 | 67.1 |
| TempFlex-Qwen3-4B (Ours) | **60.4** | 78.5 | **66.1** | **47.6** | **69.3** |

# 4 Experiments

## 4.1 Implementation Details

**Architecture.** We adopt the siglip2-so400m-patch16-naflex (Tschannen et al., 2025) model as our visual encoder, initialized with public pretrained weights. To handle longer visual sequences, we expand its original positional embedding from 256 to 2048 tokens. For the language backbone, we use the decoder-only LLM component of Gemma3-4B (Team et al., 2025) and Qwen3-4B (Yang et al., 2025a). The TFF module consists of four stacked layers with hidden size 1152. Each layer generates a $3 \times 1152$ dynamic depthwise kernel via a two-layer MLP with hidden size of 1152, encodes periodicity using five fixed frequencies {1, 2, 4, 8, 16}, and applies a linear fusion gate to predict three blending weights per patch fiber. A gated feedforward network with hidden size 2304 is appended to each layer. For patch merger, we apply a stride-2 pixel unshuffle operation to the patch embeddings, resulting in a $4\times$ reduction in sequence length.

**Training Recipe.** All models are trained on 128 NVIDIA H100 GPUs with 80GB VRAM in four distinct stages:

Table 3: Comparison with state-of-the-art methods on video benchmarks. The best results are highlighted in **bold**, and the second-best results are underlined. Most of the comparison results in the table are sourced from OpenCompass (Contributors, 2023); for entries not covered by OpenCompass, as well as all our TempFlex results, we performed a consistent evaluation using VLMEvalKit (Duan et al., 2024).

| Models | Frames | MMBench-Video | $MLVU_{val}$ | TempCompass | $LongVideoBench_{val}$ | VideoMME (wo/w sub) | |
|---|---|---|---|---|---|---|---|
| Kangaroo-8B (Liu et al., 2024b) | 64 | 1.44 | 61.1 | 62.5 | 54.8 | 56.0 | 57.6 |
| TimeMarker-8B (Chen et al., 2024d) | 128 | 1.53 | 63.9 | 60.4 | 56.3 | 57.3 | 62.8 |
| InternVL2-8B (Team, 2024) | 16 | 1.28 | 56.5 | 65.4 | 54.6 | 54.0 | 56.9 |
| MiniCPM-V-2.6-8B (Yao et al., 2024) | 64 | 1.76 | 52.5 | 57.1 | – | 60.9 | 63.6 |
| LongVA-7B (Zhang et al., 2024a) | 128 | – | 56.3 | 57.0 | 47.8 | 52.6 | 54.3 |
| Video-XL-7B (Shu et al., 2024) | 128 | – | 64.9 | – | 50.7 | 55.5 | 61.0 |
| LLava-OneVision-7B (Li et al., 2024a) | 32 | – | 64.7 | – | 56.3 | 58.2 | – |
| Qwen2-VL-7B (Wang et al., 2024a) | 64 | 1.44 | 54.8 | 59.1 | – | 59.7 | – |
| Phi4-vision-5.6B (Abdin et al., 2024b) | 64 | 0.87 | 45.6 | 50.4 | – | 42.4 | – |
| Phi3.5-vision-4B (Abdin et al., 2024a) | 16 | 1.20 | 52.5 | 59.7 | – | 51.1 | – |
| InternVL2-4B (Team, 2024) | 64 | 1.45 | 59.9 | – | 53.0 | 53.9 | 57.0 |
| InternVL2.5-4B (Chen et al., 2024e) | 64 | 1.73 | 68.3 | – | 55.2 | 62.3 | 63.6 |
| Qwen2.5-VL-3B (Bai et al., 2025) | 768 | 1.63 | 68.2 | 64.4 | 54.2 | 61.5 | 67.6 |
| TempFlex-Gemma3-4B | 128 | 1.58 | 69.0 | 63.8 | 57.2 | 63.3 | 64.3 |
| TempFlex-Qwen3-4B | 128 | **1.86** | **70.9** | **68.9** | **60.4** | **65.9** | **67.7** |

`Stage 1: Visual-Language Alignment and Encoder Adaptation.` We jointly fine-tune the visual encoder and the MLP projector, setting the learning rates to 5e-5 and 1e-3, respectively. This stage uses 13.67 million high-quality image-caption pairs with diverse resolutions to align the visual and language modalities and adapt the pretrained visual backbone.

`Stage 2: Image Instruction Fine-Tuning.` In this stage, we train the visual encoder, MLP projector, and LLM simultaneously. The visual encoder is trained with a learning rate of 2e-6, while the remaining components use 1e-5. This stage utilizes 27.94 million multi-task instruction samples spanning general question answering, OCR, document/chart/screen understanding, mathematical reasoning, and pure language tasks.

`Stage 3: Temporal Fusion Alignment.` We activate the TFF module and the MLP projector, using learning rates of 5e-4 and 1e-5, respectively. This stage is trained on 900K video-captioning pairs to guide temporal reasoning and dynamic representation learning.

`Stage 4: Video Instruction Fine-Tuning.` All model parameters are updated in this stage. The visual encoder is trained with a learning rate of 2e-6, and the rest of the model with 1e-5. The total training data includes 6.68 million samples: 2 million high-quality video instruction samples synthesized in-house, 1.4 million public video instruction-tuning samples, and 3.28 million image-based instruction samples to retain the model's image understanding ability. During training, a maximum of 64 frames is sampled per video, while 128 frames are used at test time.

For public data sourcing, we follow LLaVA-OneVision (Li et al., 2024a) and Infinity-MM (Gu et al., 2024) in constructing the instruction tuning corpus. All datasets undergo low-quality filtering, and we provide detailed dataset composition for each stage in Appendix A.3.

In terms of training settings, Stage 4 is trained for 2 epochs with a batch size of 256, while all other stages use a batch size of 512 and are trained for 1 epoch. We employ a cosine learning rate scheduler with a warmup ratio of 0.03. To enhance training efficiency and scalability, we apply DeepSpeed ZeRO-2 optimization, gradient checkpointing, and Flash Attention throughout the training process.

**Evaluation.** We conduct comprehensive evaluations on a diverse set of image and video benchmarks. For image-based tasks, we report results on MMStar (Chen et al., 2024b), MMBench (Liu et al., 2024c), RealWorldQA, Hallusionbench (Guan et al., 2024), MathVista (Lu et al., 2023), MMMU (Yue et al., 2024), AI2D (Kembhavi et al., 2016), DocVQA (Mathew et al., 2021), and TextVQA (Singh et al., 2019). For video-level evaluation, we include VideoMME (Fu et al., 2024), LongVideoBench (Wu et al., 2024a),

TempCompass (Liu et al., 2024d), MMBench-Video (Fang et al., 2024), and MLVU (Li et al., 2024b). We compute accuracy using the standardized VLMEvalKit (Duan et al., 2024) open-source evaluation framework to ensure consistency and reproducibility across different tasks.

## 4.2 Main Results

**Comparison with SoTAs in Image Understanding.** Table 2 shows that our TempFlex family sets a new performance bar for 4B-scale vision–language models (Beyer et al., 2024; Team et al., 2025; Abdin et al., 2024a;b; Bai et al., 2025; Wang et al., 2024a; Gao et al., 2024b; Chen et al., 2024e; Team, 2024) across nine image–understanding benchmarks. TempFlex-Qwen3-4B attains the highest overall average of 69.3%, surpassing the previous best 4B model, InternVL2.5-4B (Chen et al., 2024e), by 1.3 percentage points (pp) and even edging out the larger 5.6 B Phi4-vision (Abdin et al., 2024b) by 1.0 pp. This lead is driven by consistent gains on both general visual-question-answering tasks and specialist benchmarks: TempFlex-Qwen3-4B establishes new state-of-the-art results on MMStar (60.4%), RealWorldQA (66.1%), HallBench (47.6%), and MathVista$_{mini}$ (67.8%), while matching or exceeding prior bests on the OCR-centric TextVQA (79.1%) and DocVQA (91.0%). TempFlex-Gemma3-4B likewise performs strongly, posting the top result on the comprehensive reasoning benchmark MMBench-EN$_{dev}$ at 79.2% and ranking a close second to Phi4-vision (Abdin et al., 2024b) on AI2D (82.0% vs. 83.0%). It is worth emphasizing that, compared to the original Gemma3 (Team et al., 2025), TempFlex-Gemma3 elevates the overall average from 55.3% to 67.1%, marking a notable improvement of 11.8 percentage points. This gain, achieved by simply adopting a different visual-encoding strategy, underscores the strength of TempFlex's natively any-resolution visual representation.

**Comparison with SoTAs in Video Understanding.** Table 3 benchmarks our two 4B-parameter variants—TempFlex-Gemma3-4B and TempFlex-Qwen3-4B—against the strongest published video-capable MLLMs (Liu et al., 2024b; Chen et al., 2024d; Team, 2024; Yao et al., 2024; Zhang et al., 2024a; Shu et al., 2024; Li et al., 2024a; Wang et al., 2024a; Bai et al., 2025; Abdin et al., 2024b;a; Chen et al., 2024e) spanning 3B to 8B parameters. Despite their compact size, both TempFlex models either match or surpass every larger competitor on all five representative benchmarks.

TempFlex-Qwen3-4B attains the best overall performance across all benchmarks. It achieves 1.86 on MMBench-Video (Fang et al., 2024), outperforming InternVL2.5-4B (1.73) and TimeMarker-8B (1.53); 70.9 on MLVU$_{val}$ (Zhou et al., 2024), surpassing InternVL2.5-4B (68.3) and Qwen2.5-VL-3B (68.2); 60.4 on LongVideoBench$_{val}$ (Wu et al., 2024a), ahead of TimeMarker-8B and LLava-OneVision (both 56.3); and 65.9 on VideoMME (Fu et al., 2024), exceeding InternVL2.5-4B (62.3) and Kangaroo-8B (56.0). On TempCompass (Liu et al., 2024d), it reaches 68.9, the highest score to date, indicating improved temporal alignment and multimodal reasoning.

TempFlex-Gemma3-4B also shows consistently strong performance: 1.58 on MMBench-Video, 69.0 on MLVU$_{val}$, 57.2 on LongVideoBench$_{val}$, 63.3 on VideoMME, and 63.8 on TempCompass. These results outperform most 4B and even some 7B–8B baselines, such as InternVL2-4B, Phi3.5-Vision-4B, and LongVA-7B.

It is worth noting that the strong performance of TempFlex-Qwen3-4B on TempCompass further confirms the effectiveness of the proposed *Temporal Fiber Fusion* module in enhancing temporal understanding across diverse video reasoning tasks.

**Qualitative Results.** In Figure 4, we illustrate our model's exceptional visual understanding and text generation capabilities across various scenarios, encompassing both image and video. More qualitative results are provided in the Appendix A.4.

## 4.3 Ablation Study

**Effect of TempFlex encoding.** To rigorously assess the impact of the visual-encoding module, we compare TempFlex's native any-resolution strategy with two prevailing alternatives: (1) `Fix-resolution encoding` (Liu et al., 2023; Zhu et al., 2023; Li et al., 2023a), which resizes every image to a fixed spatial size before feature extraction; and (2) `Tiled-patch encoding` (Liu et al., 2024a; Li et al., 2024a; Wu et al.,

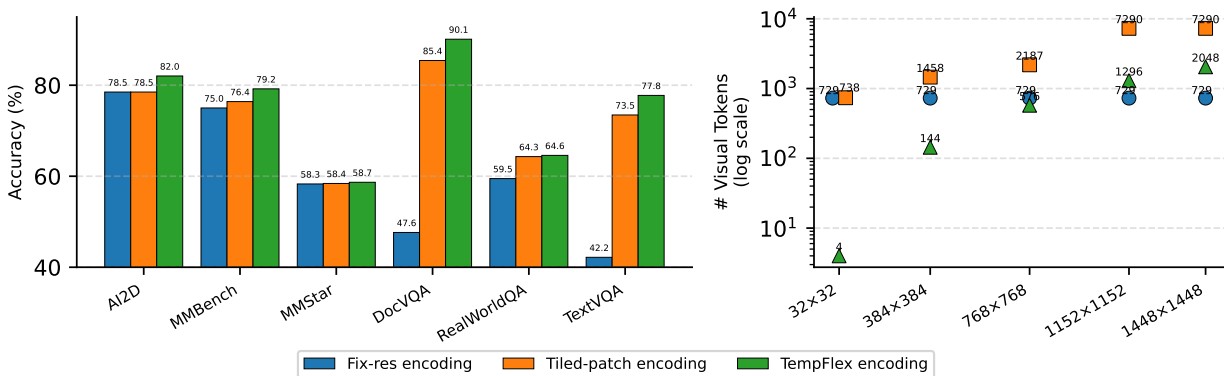

Figure 3: Comparison of performance (left) and efficiency (right) across different visual encoding strategies. Efficiency is measured by the number of visual tokens generated under varying input resolutions for each encoding method.

Table 4: Performance evaluation of Temporal Fiber Fusion module on video benchmarks. TG denotes TempFlex-Gemma3-4B, while TQ refers to TempFlex-Qwen3-4B model.

| # | Models | | | VideoMME (w/o subs) | | | | TempCompass |
|---|---|---|---|---|---|---|---|---|
| | Local fiber | Memory fiber | Periodic fiber | Overall | Short video | Medium video | Long video | |
| | | | | 57.1 | 70.9 | 54.1 | 46.2 | 58.2 |
| | ✓ | ✓ | | 58.4 | 72.2 | 55.3 | 47.7 | 59.4 |
| TG | ✓ | | ✓ | 57.9 | 71.9 | 55.0 | 46.8 | 58.8 |
| | | ✓ | ✓ | 58.2 | 72.0 | 55.1 | 47.5 | 59.1 |
| | ✓ | ✓ | ✓ | 58.8 | 72.7 | 55.7 | 47.9 | 59.7 |
| | | | | 58.4 | 72.5 | 54.6 | 48.1 | 61.7 |
| | ✓ | ✓ | | 59.2 | 73.2 | 55.8 | 48.6 | 63.0 |
| TQ | ✓ | | ✓ | 58.7 | 72.9 | 55.0 | 48.2 | 62.6 |
| | | ✓ | ✓ | 59.0 | 73.1 | 55.4 | 48.5 | 62.9 |
| | ✓ | ✓ | ✓ | 59.6 | 73.6 | 56.4 | 48.7 | 63.2 |

2024b; Team, 2024; Chen et al., 2024e), which tessellates the input into 384 × 384 crops whose features are concatenated into a long visual token sequence. All experiments are conducted under identical training settings with Gemma3 (Team et al., 2025) as the LLM backbone and the only variation is the visual encoding strategy. This ensures a fair and controlled comparison.

As shown in Figure 3 (left), TempFlex consistently outperforms both baselines across six image benchmarks. It yields +3.5 percentage points (pp) improvement on AI2D (Kembhavi et al., 2016) and +4.2 pp on MMBench (Liu et al., 2024c), while achieving comparable performance on MMStar (Chen et al., 2024b). The advantage becomes more prominent in high-resolution, detail-centric benchmarks. On DocVQA (Mathew et al., 2021), TempFlex achieves 90.1% accuracy, surpassing fix-resolution (47.6%) and tiled-patch (85.4%) by a large margin. Similarly, on TextVQA (Singh et al., 2019), TempFlex reaches 77.8%, outperforming fix-resolution (42.2%) and tiled-patch (73.5%).

Figure 3 (right) compares token efficiency across resolutions. Fix-resolution encoding, which compresses all inputs to 384x384 pixels, yields a constant number of visual tokens of 729 regardless of input size. While this approach is computationally efficient, it leads to a significant performance drop on high-resolution benchmarks such as DocVQA. Tiled-patch encoding preserves more detail but significantly increases visual token count, reaching 7,290 tokens at 1152×1152 resolution—over 5.6× higher than TempFlex at the same resolution. In contrast, our encoding strategy is significantly more efficient, aided by patch merging: producing only 4 tokens at 32×32, 576 at 768×768, and 2,048 at 1448×1448. These results highlight TempFlex's favorable accuracy–efficiency trade-off: it enables global attention over high-resolution inputs while reducing token overhead by at least 72% compared to patch-wise encoding.

**Effect of Temporal Fiber Fusion.** To validate the effectiveness of the proposed TFF module, we conducted a detailed ablation study to evaluate the impact of each of the three branches in the TFF on two TempFlex-VL vairants. All experiments were conducted on the VideoMME (Fu et al., 2024) and TempCompass (Liu et al., 2024d) benchmark. For training efficiency, we sampled 30% of the training data while keeping other experimental settings consistent to ensure fair comparability of the results. As shown in the Table 4, the checkmarks (✓) indicate the presence of each fiber branch. From the experimental results, it is evident that all three branches—local, memory, and periodic—contribute to significant performance improvements.

Table 5: Comparison of the model performance with and without synthetic data on VideoMME.

| Models | Overall | | Short Video | | Medium Video | | Long Video | |
|---|---|---|---|---|---|---|---|---|
| | w/o subs | w subs | w/o subs | w subs | w/o subs | w subs | w/o subs | w subs |
| TempFlex-VL-4B | 63.3 | 64.3 | 76.3 | 78.2 | 63.3 | 65.1 | 50.3 | 49.6 |
| w/o TempFlex-2M | 62.5 | 63.5 | 75.3 | 77.0 | 60.7 | 63.8 | 51.7 | 49.8 |

**Effects of Synthetic Data.** To assess the contribution of synthetic video-text pairs, we conduct an ablation study on the Video-MME (Fu et al., 2024) benchmark by removing the TempFlex-2M synthetic subset from the training corpus. Table 5 shows performance with and without synthetic data across different video lengths and subtitle availability.

Removing synthetic data leads to consistent drops in performance, both in the presence and absence of subtitles. The overall accuracy declines by 0.74–0.78% on average, and more notably on medium-length videos, suggesting that synthetic supervision enhances temporal reasoning.

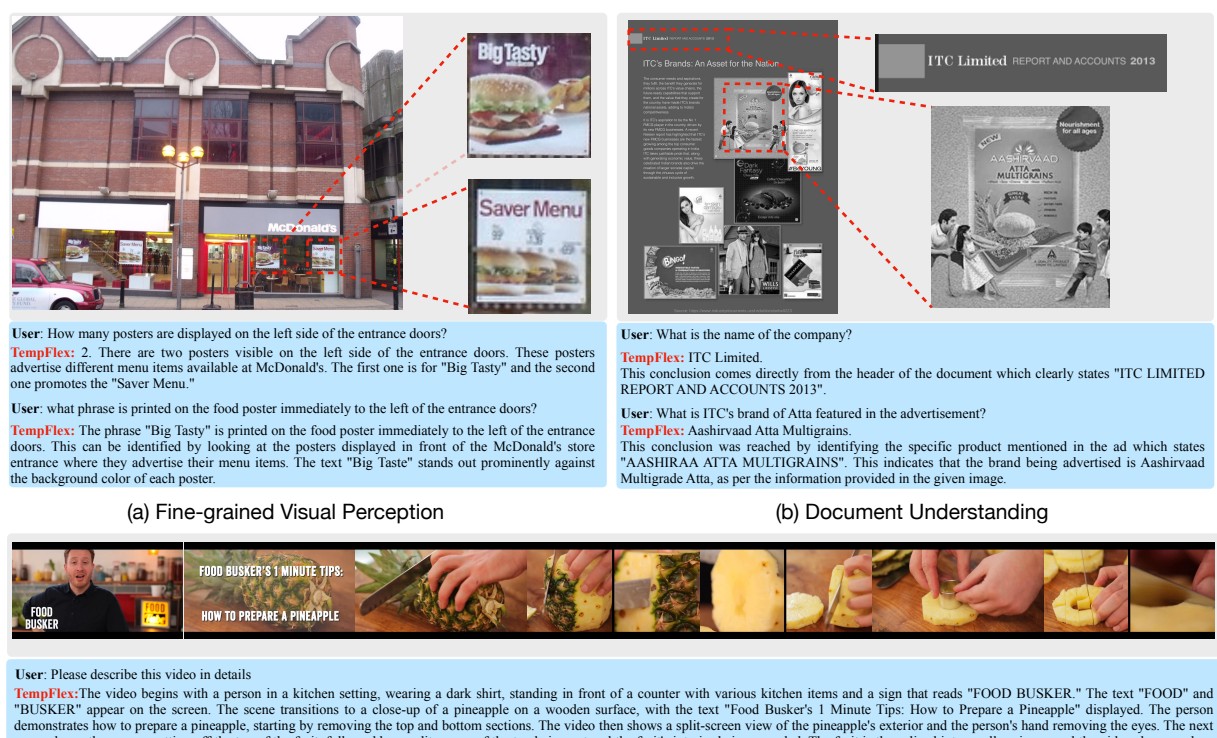

(a) Fine-grained Visual Perception

(b) Document Understanding

(c) Video Understanding: Action Recognition and Temporal Perception

Figure 4: Quantitative Results for Image and Video dialogues.

## 5 Conclusion and Limitation

This work introduces TempFlex-VL, a fully reproducible multimodal language model that overcomes two persistent bottlenecks in vision–language research: resolution rigidity and the lack of explicit temporal reasoning. By pairing a native any-resolution adaptation of SigLIP2 with a lightweight Temporal Fiber Fusion module, TempFlex-VL-4B delivers state-of-the-art performance across a broad suite of image and video benchmarks while using far fewer visual tokens than prior systems. We further release the accompanying TempFlex-2M synthetic video corpus, model weights, and training code to facilitate transparent, community-driven progress.

**Limitation** While TempFlex-VL benefits from strong supervised pre-training, it has not yet been post-trained with reinforcement-learning techniques that could better align its outputs with downstream user preferences. We leave the integration of reward modelling and preference optimisation as promising future work to push the model's capabilities beyond the current supervised ceiling.

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

# A   Appendix

Table 6: Comparison of visual encoding strategies on a $1280 \times 720$ input image.

| Method | Token Count | Distortion | Memory Cost | Feature Quality | Remarks |
|---|---|---|---|---|---|
| Fix-Resolution | 576 | ✓ | Low | Moderate | Simple but lossy due to resizing |
| Tiled Patch | $\sim 4032$ | ✗ | High | Medium | Fragmented tokens, weak cross-tile links |
| TempFlex (Ours) | $\sim 900$ | ✗ | Moderate | High | No resizing, full-resolution attention |

## A.1   Details of visual encoding process

In this section, we provide a detailed explanation of the three visual encoding strategies considered in our study—*Fix-Resolution Encoding, Tiled Patch Encoding*, and our proposed *TempFlex Encoding*. We also compare these strategies quantitatively in terms of visual token efficiency and computational overhead.

**Fix-Resolution Encoding**   This approach resizes every input image to a fixed spatial resolution $(H, W) = (384, 384)$:

$$\mathbf{I}' = \mathrm{Resize}(\mathbf{I}, 384, 384),$$

where $\mathbf{I} \in \mathbb{R}^{H_0 \times W_0 \times 3}$ is the original image. The resized image is then fed into a frozen SigLIP2 encoder:

$$\mathbf{V}_{\mathrm{fix}} = f_{\mathrm{vis}}(\mathbf{I}') \in \mathbb{R}^{576 \times D},$$

where each visual token corresponds to a $16 \times 16$ patch and $D$ is the embedding dimension. This method is simple but introduces spatial distortion when processing images with diverse aspect ratios.

**Tiled Patch Encoding**   To retain high-resolution features, the image is divided into non-overlapping tiles of size $384 \times 384$, each processed independently by the encoder:

$$\mathbf{V}^{(k)} = f_{\mathrm{vis}}(\mathbf{I}^{(k)}) \in \mathbb{R}^{576 \times D}, \quad k = 1, \ldots, K,$$

where $K$ denotes the number of tiles. Additionally, a resized global view $\mathbf{I}_{\mathrm{global}} = \mathrm{Resize}(\mathbf{I}, 384, 384)$ is encoded:

$$\mathbf{V}_{\mathrm{global}} = f_{\mathrm{vis}}(\mathbf{I}_{\mathrm{global}}) \in \mathbb{R}^{576 \times D}.$$

The final representation is the concatenation:

$$\mathbf{V}_{\mathrm{tiled}} = \mathrm{Concat}(\mathbf{V}^{(1)}, \ldots, \mathbf{V}^{(K)}, \mathbf{V}_{\mathrm{global}}) \in \mathbb{R}^{(576K + 576) \times D}.$$

While this strategy preserves detail and introduces global context, it significantly increases token length and lacks inter-tile continuity.

**TempFlex Encoding (Ours)**   Our proposed method avoids resizing and tiling by processing the input at its native resolution. Given $\mathbf{I} \in \mathbb{R}^{H_0 \times W_0 \times 3}$, we first obtain patch-level embeddings using a fixed patch size of 16:

$$\mathbf{V}_0 = f_{\mathrm{patch}}(\mathbf{I}) \in \mathbb{R}^{\left( \frac{H_0}{16} \cdot \frac{W_0}{16} \right) \times D}.$$

We then apply a pixel unshuffle operator with stride $s = 2$, effectively reducing the token count:

$$\mathbf{V}_{\mathrm{tempflex}} = \mathrm{Unshuffle}_2(\mathbf{V}_0) \in \mathbb{R}^{\left( \frac{H_0}{32} \cdot \frac{W_0}{32} \right) \times (4D)}.$$

This operation maintains spatial integrity, allows native resolution input, and significantly reduces the number of visual tokens.

**Token Efficiency and Computational Comparison**   We benchmark the encoding strategies using a representative resolution of $1280 \times 720$ and summarize their properties in Table 6.

The token count for TempFlex is approximately an order of magnitude lower than the tiled patch approach while maintaining the native spatial structure of the input. This makes it significantly more efficient and flexible for both short and long video sequences.

## A.2   Details regarding the construction of the TempFlex-2M

In this section, we provide detailed information on the construction of the TempFlex-2M dataset, supplementing the brief introduction in the main paper. As described in the main text, TempFlex-2M is a large-scale synthetic video-language instruction dataset curated to facilitate instruction tuning for multimodal large language models (MLLMs). The dataset is synthesized using GPT-4o based on four primary data sources: Virpt (Yang et al., 2024), VATEX (Wang et al., 2019), OpenVid (Nan et al., 2024), and FineVideo (Farré et al., 2024).

To ensure diversity and comprehensive video understanding, we define twelve instruction dimensions that reflect distinct reasoning or perceptual challenges, including but not limited to temporal reasoning, motion tracking, event prediction, object interaction, and spatial relationships. GPT-4o is guided to generate both caption-style and open-ended VQA-style annotations for each video segment, tailored to each of these dimensions.

Figure 5(a) illustrates the distribution of generated samples across the twelve instruction dimensions, showing a balanced and diverse composition. Figure 5(b) reports the sample statistics from each of the four data sources, separately for caption and open-ended VQA types, highlighting the multi-format coverage contributed by each source.

We further detail our prompt engineering strategies in Figures 6 and 7, where we present representative prompts used to guide GPT-4o in generating instruction-following data. Finally, Figure 8 presents a complete example from the TempFlex-2M dataset

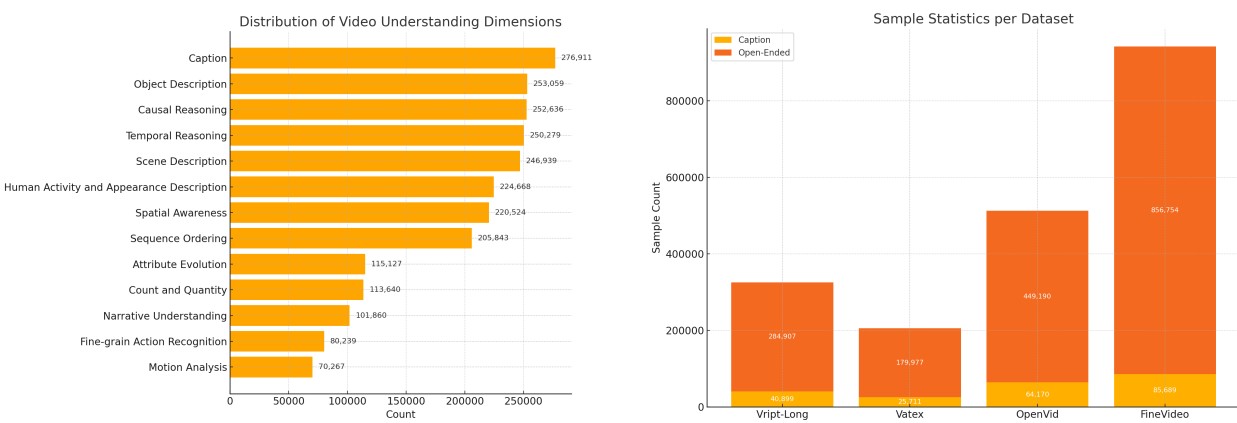

(a) Distribution of video understanding dimensions.

(b) Sample statistics per dataset.

Figure 5: Overview of video understanding tasks and dataset compositions.

## A.3   Details of the training data composition

This section supplements the training data composition details. For image-text data, we follow the data setup from Infinity-MM (Gu et al., 2024) and LLaVA-OneVision (Li et al., 2024a), while applying additional filtering to remove low-quality samples, such as those generated by GPT-4V.

For video-text data, we primarily use the LLaVA-Video-178K (Zhang et al., 2024c) dataset and our own synthetic dataset, TempFlex-2M, which provides high-quality, instruction-oriented annotations.

Table 7 summarizes the amount and composition of training data used in each stage, showing the progression from image-based to video-based multimodal supervision.

Table 7: Overview of training data composition across all stages. We categorize data into image caption, video caption, and comprehensive instruction data. Each row details the data type, total volume, and fine-grained composition per stage.

| Training Stage | Data Category | Size | Data Composition |
|---|---|---|---|
| #1 Alignment and Adaptation | Image Caption Data | 13.67M | General Instruction Data 13.67M |
| #2 Image Instruction Tuning | Comprehensive Data | 27.94M | General Instruction Data 9.24M |
| | | | OCR Data 2.6M |
| | | | Doc/Chart/Screen Data 5.8M |
| | | | Math/Reasoning Data 1.3M |
| | | | Text Instruction Data 9M |
| #3 Temporal Fusion Alignment | Video Caption Data | 900K | General Instruction Data 900K |
| #4 Video Instruction Tuning | Comprehensive Data | 6.68M | General Instruction Data 1.3M |
| | | | OCR Data 0.34M |
| | | | Doc/Chart/Screen Data 0.1M |
| | | | Math/Reasoning Data 0.14M |
| | | | Synthetic data 1.4M |
| | | | Public llava-video-178k 1.4M |
| | | | Our TempFlex Video Data 2M |

### A.4 More Qualitative Results

In this section, we provide additional qualitative case studies to further demonstrate the capabilities of our models. Specifically, we present results from both TempFlex-Gemma3-4B and TempFlex-Qwen-4B across a diverse set of multimodal tasks.

The examples cover a wide range of scenarios, including image understanding 9 10, optical character recognition (OCR) 11, document and chart comprehension 12, and video-based instruction following 13 14. These cases illustrate the generalization ability and robustness of our models across both static and temporal visual inputs.

Through these qualitative results, we aim to provide a more comprehensive view of the strengths of the TempFlex framework in handling complex and diverse multimodal tasks.

**GPT4o's Prompt for Constructing Video Instruction Tuning dataset**

You are an AI assistant tasked with generating questions and answers about video content to create a video instruction tuning dataset.

###TASK:
1. Users will provide multiple stitched images, each containing 9 frames. These frames are extracted from the original video at a frame rate of 1 fps and are arranged from left to right and top to bottom in tic-tac-toe style(3x3). The order of the squares is as follows: the first row from left to right represents 1, 2, 3; the second row from left to right represents 4, 5, 6; and the third row from left to right represents 7, 8, 9. Each row of images is sequentially coherent in terms of time from left to right. Multiple stitched images are also coherent over time. For example, the 9 frames in the first stitched image represent frames from 0 to 8 seconds, while the 9 frames in the second stitched image represent frames from 9 to 17 seconds. Please note that any white images within the stitched images are meaningless and can be ignored.
2. Users will also provide 1. ASR text corresponding to the video; 2. Activities happening in the video; 3. Objects / Props that appear in the video; 4. Video brief summary;
3. Generate a detailed and accurate description of a video based on multiple stitched images and clip captions provided. Instructions for writing the detailed description: Focus on describing key visual details such as appearance, motion, sequence of actions, objects involved, and interactions between elements in the video. Leave out any descriptions about the atmosphere, mood, style, aesthetics, proficiency, or emotional tone of the video. Make sure the description is no more than 20 sentences. Combine and organize information from all frames into one clear and detailed description, removing any repeated or conflicting details. Emphasize important points like the order of events, appearance and actions of people or objects, and any significant changes or movements.
4. Generate two different types of VQA: vqa-type1: VQA based solely on video content; vqa-type2: VQA combining video content and video ASR text information. All question-answer pairs should be faithful to the content of the video description and developed from different dimensions to promote comprehensive understanding of the video.

Here are some question dimensions and their explanations and exampled question-answer pairs for reference:
{**task_definitions**}

#### Guidelines For Question-Answer Pairs Generation:
- Read the stitched frames provided carefully, paying attention to the content, such as the scene where the video takes place, the main characters and their behaviors, and the development of the events.
- Generate 5 vqa-type1 question-answer pairs: VQA based solely on (1)Stitched images and (2)Props that appear in the video, neither the Answer nor the Question should contain any inferences that require ASR text or Activities.
- Generate 5 vqa-type2 question-answer pairs: VQA combining (1) Stitched images; (2)ASR text corresponding to the video; (3)Activities happening in the video; (4)Props that appear in the video; (5)Video brief summary. This means you can use all the information provided by the user to generate the VQA.
- All question-answer pairs should be faithful to the content of the video description and developed from different dimensions to promote comprehensive understanding of the video. The question-answer pairs should cover as many question dimensions and not deviate from the content of the video.
- Each VQA type generate 5 question-answer pairs, total 10 question-answer pairs. You need to select the most suitable 5 dimensions out of 15 question dimensions for this video, and generate questions based on these dimensions.
- How to choose question dimensions: Evaluate which dimensions are suitable for generating questions based on the visual content of the video. Ensure the accuracy of the generated Questions and Answers as much as possible. Prioritize selecting dimensions where both Questions and Answers are clearly demonstrated in the video. Avoid choosing dimensions that cannot be directly displayed by the video visuals and require further inference.
- The question should be objective and avoid using subjective words like infer, suggest, likely, indicate, convey, and so on.
- The questions must be complex and require at least 2 steps of thinking to reach the correct answer. A complex question does not mean a question asking combination of elements.
- Questions from different dimensions should be as distinct as possible, meaning both the questions and answers need to be diverse, rather than repeatedly asking about a specific action from different dimensions.

### Output Format:
1. Your output should be formed in a JSON file.
2. Only provide the Python dictionary string.

Figure 6: GPT4o's Prompt for Constructing TempFlex-2M Video Instruction Tuning dataset

---

**Task Definition for Generating Video Instruction Tuning Data Using GPT-4o**

TASKS = """
# Temporal Reasoning: Designed to assess reasoning about temporal relationships between actions/events. Questions involve previous, present, or next actions.
## Question-1: What activity did the child perform just before deciding to sit on the baby chair?
## Answer-1: The child was playing with a toy.

# Spatial Awareness: Tests the ability to perceive spatial relationships between observed instances in a video scene.
## Question-1: Where were the objects located in relation to the main character before they moved them?
## Answer-1: The objects were initially placed on the left side of the table next to the window.

# Causal Reasoning: Focuses on explaining actions/events, determining intentions of actions or causes for subsequent events.
## Question-1: What prompted the person to run towards the door suddenly?
## Answer-1: The person heard a loud noise coming from outside the door.

# Motion Analysis: Involves discerning variations in speed, including absolute and relative speeds.
## Question-1: How did the speed of the car change from the beginning to the end of the chase scene?
## Answer-1: The car accelerated rapidly at first but slowed down as it approached the crowded intersection.

# Binary Questions: Involves yes or no questions related to the video content.
## Question-1: Did the child look at the window before moving to the chair? (from a scene where this detail might be subtle)
## Answer-1: Yes.
## Question-2: Was there a noticeable change in lighting before the action scene started?
## Answer-2: No.

# Count and Quantity: Tests the ability to count instances of objects, people, actions, and to distinguish between old and new elements in a scene.
## Question-1: How many books did the child pick up from the floor throughout the video?
## Answer-1: The child picked up three books from the floor.

# Narrative Understanding: Challenges the ability to interpret the plot in the video.
## Question-1: What is the underlying reason for the character's behavior throughout the video?
## Answer-1: The character is trying to find a hidden object that is crucial to the story.

# Object Description: Assesses the ability to describe attributes of objects, like their appearance and function.
## Question-1: Describe the appearance and possible function of the object placed next to the lamp.
## Answer-1: The object is a spherical device with a digital screen, likely used as an alarm clock.

# Sequence Ordering: Challenges recognition of the temporal sequence of activities in videos.
## Question-1: Arrange the following events in the order they occurred: the child playing with the toy, the child sitting on the chair, the child picking up the book.
## Answer-1: The child was playing with the toy, then picked up the book, and finally sat on the chair.

# Fine-grain Action Recognition: Creates questions challenging comprehension of subtle actions.
## Question-1: What specific gestures did the character use to communicate with the other person?
## Answer-1: The character used hand signals and nodding to communicate without speaking.

# Human Activity and Appearance Description: Involves describing actions or attributes of people, such as their activities and appearances.
## Question-1: Describe the attire and actions of the main character during the garden scene.
## Answer-1: The main character is wearing a blue shirt and jeans, and they are watering the plants.

# Attribute Evolution: Centers on how attributes of objects or the entire video change over time, like size, shape, color, and more.
## Question-1: How did the color of the sky change from the beginning to the end of the scene?
## Answer-1: The sky changed from a bright blue to a deep orange as the sun set.

# Camera Movement Direction: Focuses on the direction of camera movement.
## Question-1: In which direction did the camera pan during the opening scene?
## Answer-1: The camera panned from left to right during the opening scene.

# Object Movement Direction: Emphasizes perception of object movement direction.
## Question-1: Which direction did the car move when entering the frame?
## Answer-1: The car moved from the right to the left when entering the frame.

# Scene Description: Assesses the ability to describe the major scene of the video, like where it takes place and the overall environment.
## Question-1: Describe the setting and atmosphere of the final battle scene.
## Answer-1: The final battle took place in a dimly lit warehouse filled with crates, creating a tense and gritty atmosphere.
"""

Figure 7: Task Definition for Generating TempFlex-2M Video Instruction Tuning Data Using GPT-4o

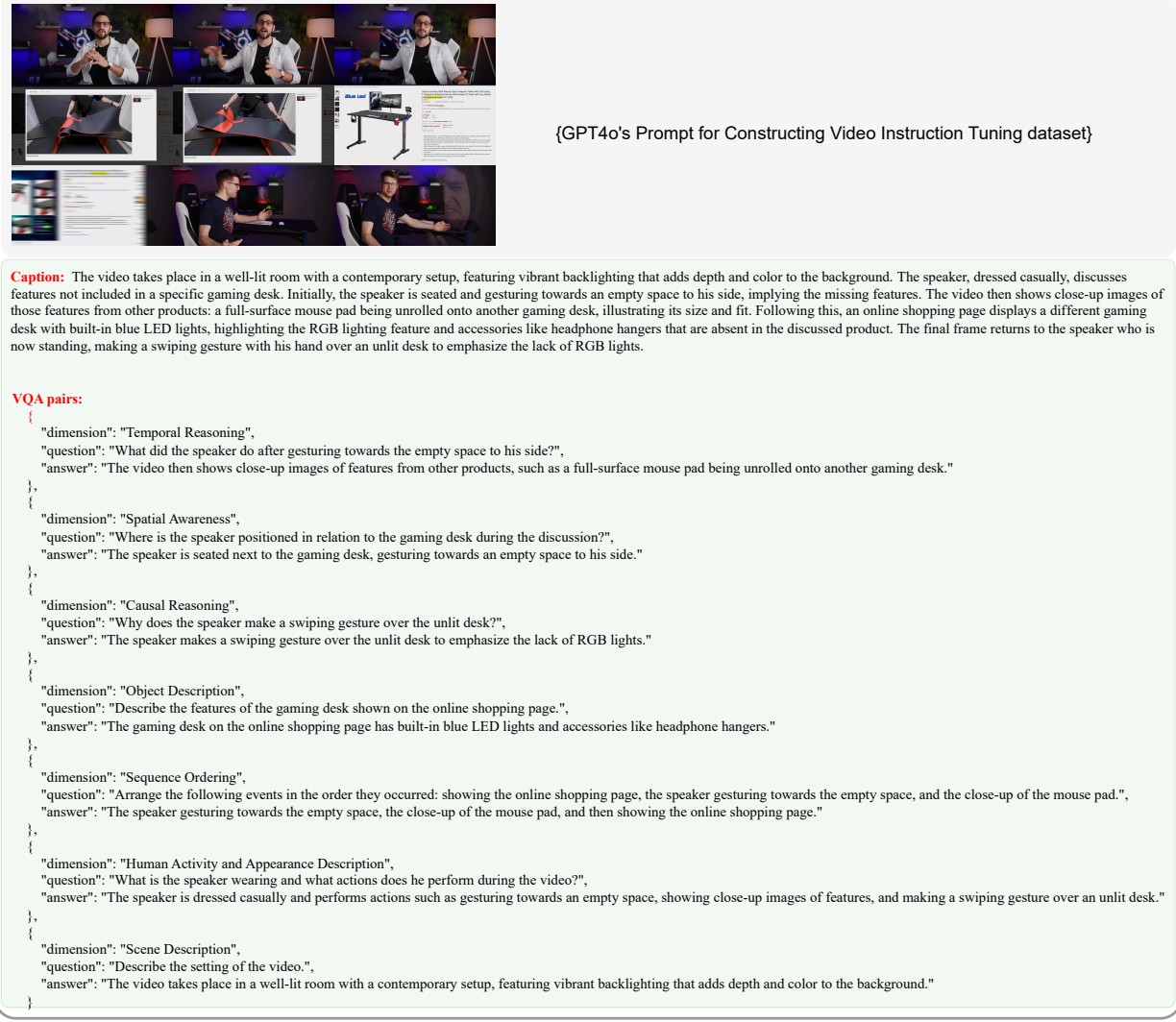

Figure 8: An example from the TempFlex-2M dataset. The top part of the image shows the input to GPT-4o, including a 3×3 grid of 9 concatenated frames along with the prompt illustrated in Figure 6. The bottom part of the image presents the output, consisting of a caption and several open-ended VQA pairs.

TempFlex-Gemma3-4B

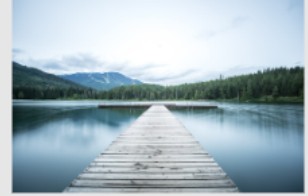 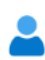

Please describe this image in details. 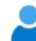

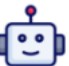

The image depicts a tranquil and serene scene of a wooden pier extending into a calm body of water, which appears to be a lake or a wide river. The pier is constructed in a straight line, with wood planks arranged in parallel, and it ends with two small floating platforms. The water is remarkably still, reflecting the surrounding environment with crystal clarity.

In the background, there are two distinct mountain ranges visible, implying that the pier is located in a valley surrounded by mountains. The mountains are covered with a mix of coniferous and deciduous trees. The mountains are partially obscured by clouds or mist, giving them a soft, hazy outline. The sky above is overcast, with clouds casting a diffused light over the entire scene, enhancing the peaceful and serene atmosphere.

The foreground is dominated by the pier, which leads the viewer's eye towards the center of the image, emphasizing the calm and stillness of the water. There are no signs of wildlife or human activity, accentuating the natural and untouched quality of the landscape.

Given the absence of any human presence and the pristine condition of the pier, it can be inferred that this location might be used for recreational purposes such as fishing, boating, or simply enjoying a quiet moment in nature. The location's natural beauty and tranquility make it an ideal setting for relaxation and reflection.

In summary, the image showcases a long wooden pier leading into a calm lake or river, surrounded by distant mountains and a cloudy sky. The reflection of the landscape on the water and the absence of human activity emphasize the location's serene and natural appeal, making it a perfect spot for relaxation and enjoying the outdoors.

Figure 9: Qualitative result of TempFlex-Gemma3-4B on image understanding.

TempFlex-Qwen3-4B

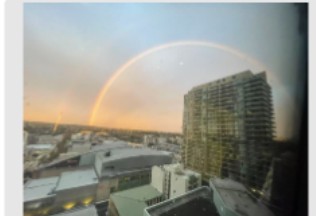 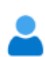

Please describe this image in details. 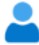

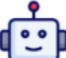

The image features a picturesque urban scene observed from a high vantage point. Dominating the upper part of the image is a prominent double rainbow. The rainbow curves gracefully from the left to the right of the frame, with the outer arc displaying a vibrant array of colors that transition from red to violet. The inner arc is less distinct and more subdued in color.

Below the rainbow, the cityscape unfolds with a multitude of buildings. The foreground is occupied by a tall, multi-story residential building with numerous balconies, some of which are visible due to the perspective. The building's facade is a mix of light and dark shades, creating a textured appearance.

In the midground, various other buildings can be seen. These buildings vary in height and design, with some appearing to be commercial or office spaces due to their flat rooftops and the presence of signage. The signs include texts such as "Westfield," indicating the presence of a commercial district or shopping area.

The background extends further into the horizon, where more urban structures are faintly visible, blending into the sky. The sky itself is a gradient of colors, suggesting either a sunrise or sunset, which complements the vibrant hues of the rainbow. A crane is visible on the left side of the image, indicating ongoing construction or development within the city. The presence of the crane adds a dynamic element to the otherwise static urban landscape.

The overall composition of the image captures the juxtaposition of natural beauty (the rainbow) and urban development, creating a visually striking scene that highlights the coexistence of nature and man-made structures in a modern city.

Figure 10: Qualitative result of TempFlex-Qwen3-4B on image understanding.

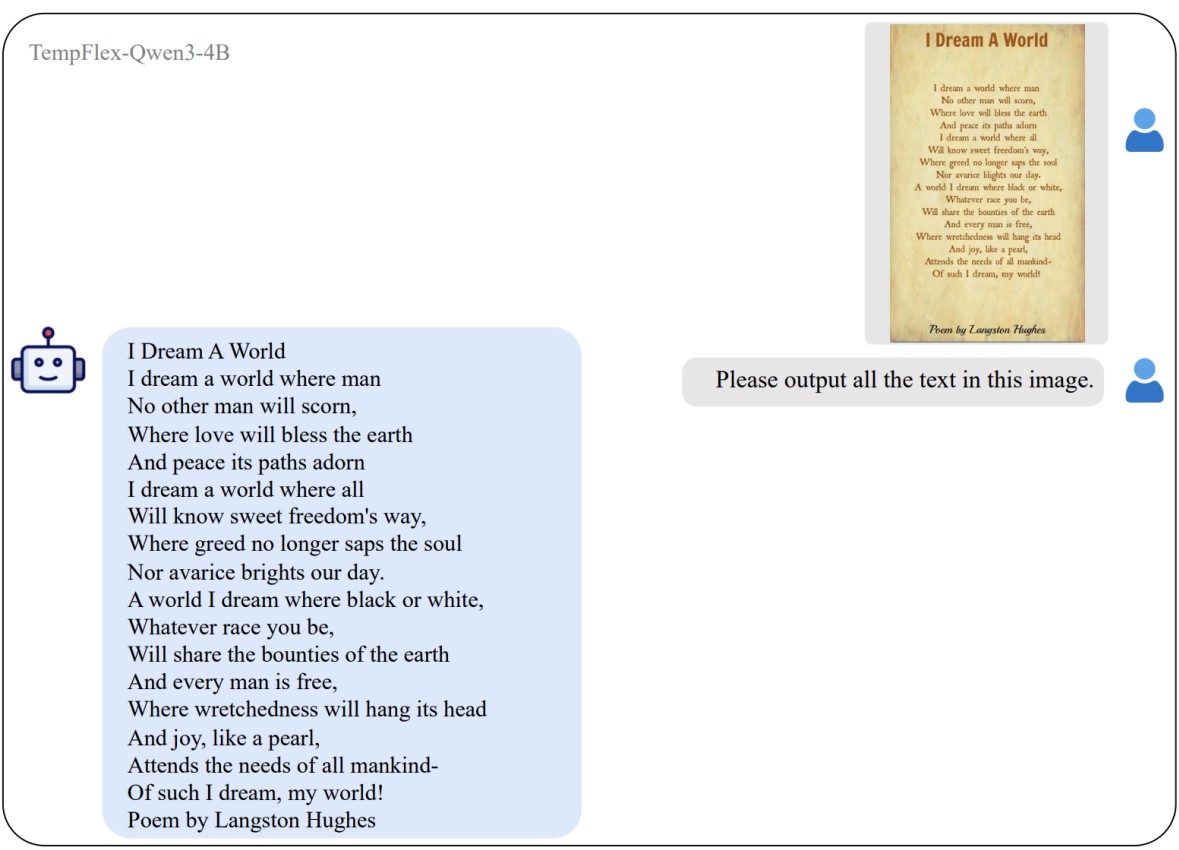

Figure 11: Qualitative result of TempFlex-Qwen3-4B on OCR.

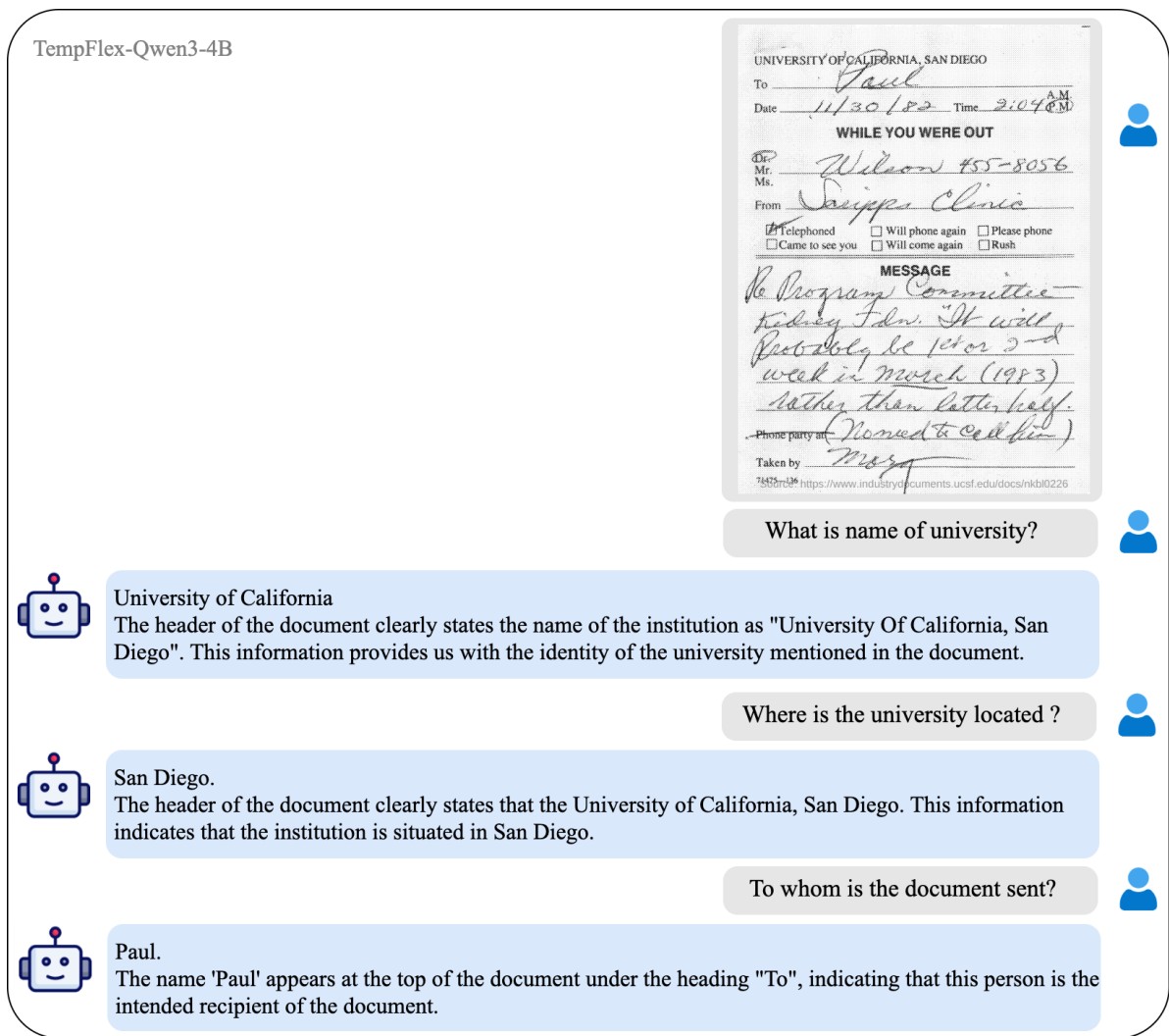

Figure 12: Qualitative result of TempFlex-Qwen3-4B on Document understanding

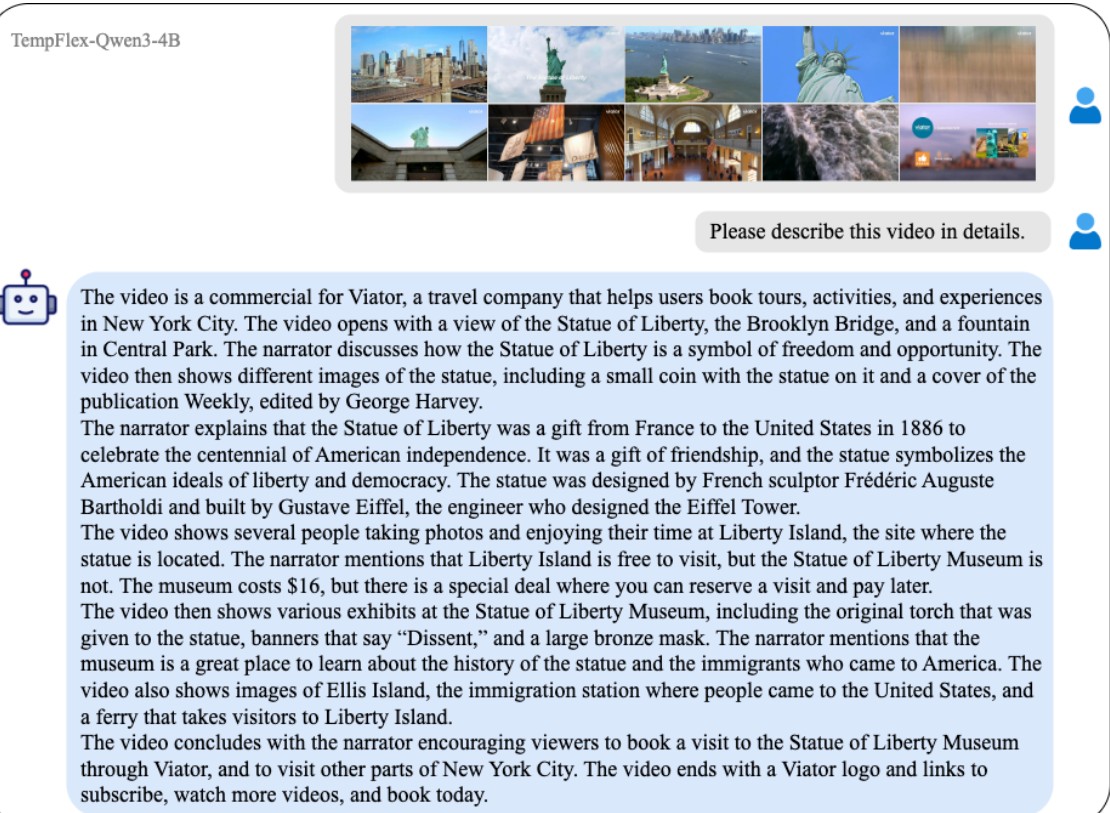

Figure 13: Qualitative result of TempFlex-Gemma3-4B on Video understanding

Figure 14: Qualitative result of TempFlex-Qwen3-4B on Video understanding

