# OpenReview forum: "TempFlex: Advancing MLLMs with Temporal Perception and Natively Scalable Resolution Encoding"
_TMLR — Accepted by TMLR_

### Review · Reviewer_Y7r4 · 2025-08-26

**Summary Of Contributions:**

This paper proposes a MLLM called TempFlex for improving both efficiency and temporal reasoning in vision-language training tasks. The model supports any-resolution visual encoding with a temporal fiber fusion module. The paper also constructs a TempFlex-2M dataset that provides richer temporally aligned supervision. The performance of the model is demonstrated via comprehensive empirical studies.

Some potential weaknesses:
1. While the dataset boosts performance, it is synthetic and may not be generalizable to real-world scenarios.
2. Training can be more costly when compared to existing models.
3. The TTF module introduce significant model complexity, which may complicate users integrating it into other pipeline. Though the paper is well-written, but it can be challenging for readers not familiar with the field as the description of the pipeline is brief.

**Additional Comments:**

NA

**Audience:**

Yes

**Audience Explanation:**

The paper’s findings would interest TMLR’s audience, particularly those working on multimodal large language models, video understanding, and efficiency in model design. The proposed any-resolution encoding, lightweight temporal fusion module, and large-scale synthetic dataset address well-known bottlenecks and demonstrate strong empirical gains, making the work relevant to both researchers and practitioners.

**Claims And Evidence:**

Yes

**Claims Explanation:**

1. The evidence of "any-resolution visual encoding reduces tokens while preserving performance" is clear and convincing, with support from solid empirical evidence.
2. The improvement in temporal reasoning via TFF is also strongly supported and well-structured.
3. The generated dataset is demonstrated to enhance video-language pretraining.
4. TempFlex is slightly outperforming the state-of-the-art approaches.

**Requested Changes:**

1. Describe the pipeline in more details, with notation and terminologies clearly explained.
2. Provide more insight on how the generated dataset contributes to real-world tasks.

---

> ### Author Response · Authors · 2025-10-11
> **Response to Reviewer Y7r4**
>
> We thank Reviewer Y7r4 for the valuable and insightful feedback, as well as for the quick review! We are pleased the reviewer acknowledged our model’s efficiency and temporal reasoning improvements and are happy to address the concerns.
>
> **Weakness:**
>
> **W1:Generalizability of the synthetic dataset**
> Thank you for the thoughtful comment. We agree that assessing the generalization ability of synthetic data is essential.
>
> To ensure TempFlex-2M generalizes well to real-world scenarios, we grounded it in authentic video content rather than generating it entirely from scratch. Specifically, all video sources are drawn from real-world datasets like FineVideo, Virpt, and VATEX. The “synthetic” nature of TempFlex-2M lies primarily in data augmentation, not in content fabrication. We used carefully designed prompts (Fig. 6) to guide GPT-4o in generating VQA pairs that are faithful to the original video content, enforcing semantic consistency while increasing linguistic diversity.
>
> To verify its real-world generalization, we further evaluated models trained on TempFlex-2M on the Video-MME benchmark, which uses authentic videos and human-annotated QA pairs. As shown in Table 5, models trained with TempFlex-2M achieve significant improvements on Video-MME, demonstrating that the augmented data effectively enhance model performance beyond the synthetic domain.
>
> Our goal is not to replace real datasets but to complement them through a realism-grounded and fidelity-preserving augmentation strategy. We have clarified this intent in the revised manuscript. We also believe synthetic data are crucial for advancing Video-MLLMs, as they enable controllable and scalable instruction generation, which is often infeasible with human-annotated datasets that are typically limited to simple caption pairs.
>
> **W2:Potentially higher training cost**
> Thank you for the comment. We understand the concern regarding the potential increase in training cost. However, the additional overhead introduced by the proposed Temporal Fiber Fusion (TFF) module is minimal from both parameter and computational perspectives.
>
> Parameter efficiency: TFF is designed as a lightweight and plug-and-play component, introducing only 36M parameters, which accounts for merely 0.8% of the total model size.
>
> Training efficiency: Although integrating TFF may slightly increase the training time, the dominant factor affecting efficiency is the number of visual tokens. Compared to the tiled-patch encoding used in InternVL2.5-4B, our TempFlex encoding reduces visual tokens by over 72%, resulting in approximately 1.6× higher training throughput overall.
>
> Inference efficiency: Unlike attention-based temporal modules with O(N²D) complexity, TFF adopts a more efficient O(ND) formulation. This leads to substantially lower FLOPs, memory consumption, and latency, particularly in long-sequence or video scenarios.
>
> We will include a more detailed cost and efficiency analysis in the revised manuscript.
>
> **W3:Complexity and integration of TFF**
> Sorry for the confusion. In response, we have revised Section 3.2 and provided a more comprehensive description of the overall pipeline, including clearer explanations to help readers better understand the design and workflow (see R1).
>
> Furthermore, we will release the complete training code along with examples demonstrating how to integrate the TFF module into other pipelines, enabling users to easily reproduce our results and adapt the module for their own applications.
>
> **Requested Changes:**
>
> **R1: Describe the pipeline in more details, with notation and terminologies clearly explained.**
> Thank you for the valuable suggestion. We have revised the paper to include a more detailed description of the overall pipeline, with emphasis on the TFF module. Specifically, in Section 3.1 (Network Architecture), we clarify the relationships among the Visual Encoder, TFF Module, PatchMerger, and the Large Language Model. In Section 3.2 (TFF), we provide a comprehensive overview of the module’s three branches — the Local Path, Memory Path, and Periodic Path — and describe how they interact to model temporal dynamics. Furthermore, in Section 3.2.1 (Module Formulation), we present more precise mathematical definitions, including the associated notations and terminologies for each branch.
>
> **R2: Provide more insight on how the generated dataset contributes to real-world tasks.**
> Thank you for this valuable suggestion. As noted in our response to W1, we agree that clarifying the real-world relevance of the generated dataset is important. Accordingly, we will expand Section 3.3 to provide additional discussion and examples illustrating how the TempFlex-2M dataset enhances temporal reasoning and transferability to real-world multimodal understanding tasks.
>
> Thank you for your valuable suggestions! We hope we could address your concerns satisfactorily. Please let us know if there are any new concerns or additional questions we can respond to.

---

### Review · Reviewer_vnUU · 2025-09-24

**Summary Of Contributions:**

This paper introduces a lightweight architecture to tackle two core limitations of current MLLMs. For image encoding, it replaces tiled patching with a natively scalable, any-resolution encoder that preserves global structure. For video encoding, it proposes the Temporal Fiber Fusion (TFF) module, which jointly models short-term motion, long-term memory, and periodic patterns through adaptive fusion.  Extensive evaluations show that the proposed approach achieves state-of-the-art or competitive performance on both image and video benchmarks, with significant gains in token efficiency.

**Audience:**

Yes

**Audience Explanation:**

he paper addresses a standing challenge in multimodal LLMs—efficient image encoding and explicit temporal modeling for video—by introducing a relatively simple yet effective architectural addition. This makes the work both accessible and impactful. Many readers would be interested in these findings, as they demonstrate how lightweight modifications can lead to meaningful gains in performance and efficiency on image and video tasks, which remain central problems in the field.

**Claims And Evidence:**

Yes

**Claims Explanation:**

The paper presents a clear problem statement, identifying the shortcomings of existing image and video encoding approaches in MLLMs. It then provides a well-elaborated description of the proposed model structure, explaining how the any-resolution encoder and the Temporal Fiber Fusion module directly target these issues. The claims are further supported by extensive experiments that are carefully designed to highlight the specific advantages of the proposed architecture. The results consistently demonstrate solid improvements over strong baselines across multiple benchmarks, which makes the evidence both convincing and credible.

**Requested Changes:**

1.  for evaluation part, I recommend that the authors add a few sentences on the evaluation metrics used and a bit more context in the main text for readers without prior knowledge to be more easily understand the numbers in tables.

2.The paper emphasizes token efficiency, but more explicit analysis of how accuracy scales with different token budgets or input resolutions would make the contribution clearer for practitioners.

3.While TFF is lightweight and effective, it would be useful to discuss or empirically compare against attention-based temporal modules. Such structures also retain memory and can capture dependencies with relatively low computation, offering an informative baseline.

---

> ### Author Response · Authors · 2025-10-11
> **Response to Reviewer vnUU**
>
> We sincerely appreciate Reviewer vnUU’s time and thoughtful feedback on our paper. We are delighted that the reviewer recognized the clarity, convincing evidence, and practical impact of our work on efficient MLLMs. We are happy to have the opportunity to address the reviewer’s questions and concerns.
>
> **Requested Changes:**
>
> **R1. Clarify evaluation metrics and provide more context**
> Thank you for your helpful suggestion. We have provided more detailed explanations of the evaluation metrics in Section 4.1 (Evaluation). Specifically, accuracy is computed using the standardized VLMEvalKit open-source framework to ensure consistency and reproducibility across tasks. All metrics are reported as percentages, calculated as the ratio of correctly answered questions to the total number of questions. For each benchmark, we strictly follow the official evaluation protocols and use the data splits defined in the original datasets.
>
> Additionally, we have clarified the notation in the tables for the VideoMME benchmarks: “w/ sub” and “w/o sub” indicate whether subtitle information is included during evaluation.
>
> **R2. Analyze how accuracy scales with token budget**
> Thank you for the valuable suggestion. Following your advice, we have added new experiments to analyze how accuracy scales with different token budgets using the TempFlex encoding strategy. The analysis is conducted on two representative benchmarks: MMStar, a general visual question answering task, and DocVQA, an OCR-related understanding benchmark focusing on high-resolution visual reasoning.
>
> The results are summarized below:
>
> | Token Budget | MMStar | DocVQA |
> | --- | --- | --- |
> | 128 | 48.2 | 47.6 |
> | 256 | 52.3 | 54.8 |
> | 512 | 56.2 | 69.5 |
> | 1024 | 57.6 | 81.4 |
> | 2048 | 58.7 | 90.1 |
>
> As shown, accuracy consistently improves with larger token budgets for both benchmarks. The gain on MMStar is relatively modest (only 1.1% drop when reducing the token budget from 2048 to 1024), indicating that TempFlex effectively preserves performance even under limited token budgets for general understanding tasks. In contrast, DocVQA shows a substantial performance gap, highlighting that sufficient visual tokens are crucial for OCR-related tasks.
>
> We have incorporated these findings and corresponding analyses into the Experiments section of the revised manuscript.
>
> **R3. Compare TFF with attention-based temporal modules**
> Thank you for the valuable suggestion.  To provide a comparison, we have added experiments using attention-based temporal modules. Specifically, following the design of TimeSformer, we applied Transformer layers along the temporal dimension on top of the visual encoder outputs. For a fair comparison, we adopted four Transformer layers, consistent with the configuration used in the TFF module. The model was trained under the same Stage 3 setup and evaluated on the Video-MME benchmark. The results are summarized below:
>
> | Models                   | **Overall**  |            | **Short Video** |            | **Medium Video** |            | **Long Video** |            |
> | :--------------------------- | :----------: | :--------: | :-------------: | :--------: | :--------------: | :--------: | :------------: | :--------: |
> |                              | **w/o subs** | **w subs** | **w/o subs**    | **w subs** | **w/o subs**     | **w subs** | **w/o subs**   | **w subs** |
> | TempFlex-Gemma3-4B (TFF)         | 63\.3        | 64\.3      | 76\.3           | 78\.2      | 63\.3            | 65\.1      | 50\.3          | 49\.6      |
> | TempFlex-Gemma3-4B (attention) | 62\.9        | 63\.8      | 76\.6           | 78\.3      | 63\.0            | 64\.7      | 49\.1          | 48\.4      |
>
> As shown, TFF shows better overall performance than the attention-based temporal module. It is slightly inferior on short videos but clearly outperforms on medium- and long-duration videos, likely because its dynamic temporal modeling allows it to capture long-range dependencies more effectively. We will include this comparison and analysis in the revised paper.
>
>
> Thank you for your valuable suggestions! We hope we could address your concerns satisfactorily. Please let us know if there are any new concerns or additional questions we can respond to.

---

### Review · Reviewer_nkzC · 2025-10-02

**Summary Of Contributions:**

This paper addresses two core limitations of existing MLLMs: excessive visual tokens with broken global context in image processing, and lack of temporal reasoning in video understanding. To this end, this paper presents TempFlex-VL, a token-efficient and temporally aware multimodal large language model (MLLM). TempFlex-VL has two key innovations: (1) a resolution-agnostic visual encoder (based on SigLIP2) that processes full high-resolution images directly without tiling, preserving global context while reducing visual tokens by at least 72% compared to tiled patch encoding; (2) a plug-and-play Temporal Fiber Fusion (TFF) module with three complementary pathways (local convolution for fine-grained motion, gated memory for long-term dependencies, periodic encoder for cyclic patterns) to capture temporal dynamics in videos. Additionally, the authors curate TempFlex-2M, a high-quality synthetic video-text corpus generated via single-stage GPT-4o visual prompting, to support large-scale video-language pretraining. Instantiated with Gemma3-4B and Qwen3-4B backbones, TempFlex-VL achieves state-of-the-art or competitive performance on diverse image (e.g., MMStar, DocVQA) and video (e.g., VideoMME, TempCompass) benchmarks, with plans to release code, models, and data publicly.

Strengths:

1. The problem to address in this work is critical. This work helps eliminate tiling for high-resolution images as well as filling the gap of explicit temporal modeling with a lightweight, modular design, avoiding the computational overhead of heavy video-specific architectures.

2. Along with the proposed approach, this paper also introduces a synthetic dataset to addresses the scarcity of diverse video-language data by adopting a single-stage "Video+Text → Caption+VQA" pipeline. The dataset has a broad coverage of 12 reasoning skills (e.g., causal reasoning, motion analysis) for comprehensive video-language pre-training.

3. The authors conduct rigorous experiments, including head-to-head comparisons with SOTA models (e.g., InternVL2.5-4B, Qwen2.5-VL-3B), ablation studies for the TFF module (validating each pathway’s contribution) and encoding strategy (token efficiency vs. accuracy), and qualitative results (image description, OCR, video understanding). This multi-faceted evaluation strongly supports the claims of the proposed innovations.

Weaknesses:

1. The design choice of the propose framework lacks justification. For example, the paper highlights that TempFlex-2M uses a single-stage pipeline to avoid information loss from multi-stage captioning, but it does not explicitly compare TempFlex-2M with a "multi-stage variant" (e.g., generating captions first, then VQA) using the same video sources. Moreover, the paper mentions using up to 128 frames at test time, but it does not discuss how TempFlex-VL performs with extremely long videos (e.g., hour-scale content, as targeted by Video-XL).

2. The comparison of inference latency is missing. The analysis only considers token efficiency. However, it is not clear the inference latency (e.g., per-image/video processing time on a standard GPU) compared to baselines like InternVL2.5-4B or Qwen2.5-VL-3B.

Minor: The presentation can be improved.

a. There are too many references in the first sentence of the introduction, which spans 6 lines.

b. The terms used are inconsistent: this paper sometimes employs MLLM and other times uses VLMs.

c. Abbreviations are defined in multiple times, e.g., Temporal Fiber Fusion (TFF).

d. Figure 4 can be moved before the conclusion section.

**Audience:**

Yes

**Audience Explanation:**

The topic studied in this paper, i.e., the scalability and efficiency of MLLMs, would be of interest to a large group of audiences.

**Claims And Evidence:**

Yes

**Claims Explanation:**

The claims made in the submission are supported by comprehensive experiments (head-to-head comparisons with SOTA models on diverse image/video benchmarks), detailed ablation studies (validating the TFF module and resolution-agnostic encoder), a high-quality synthetic dataset (TempFlex-2M) with transparent construction, and standardized evaluation via VLMEvalKit, along with qualitative results demonstrating the model’s capabilities in image description, OCR, and video understanding .

**Requested Changes:**

1. Clarify the design choice of the proposed framework.
2. Elaborate the comparison of the latency.
3. Improve the presentation quality.

---

> ### Author Response · Authors · 2025-10-11
> **Response to Reviewer nkzC**
>
> We would like to thank Reviewer nkzC for the constructive and detailed feedback. We are pleased that the reviewer recognized the importance of our work and appreciated its lightweight, modular design and the strong experimental support.  We are happy to have the opportunity to address the reviewer’s questions and concerns below.
>
> **Weaknesses:**
>
> **W1. Clarify the design choice of the proposed framework**
> - **On the single-stage pipeline**
>
>   Thank you for your valuable comment. Due to the large scale of TempFlex-2M and the high cost of generating synthetic video data, we did not conduct a direct comparison with a multi-stage variant. Instead, we allocated our resources to produce a larger quantity of single-stage synthetic samples, which provided broader coverage of diverse temporal and spatial scenarios.
>
>   During the development of our synthetic data framework, we observed that two-stage generation approaches—e.g., first generating captions and then performing VQA—tend to lose crucial temporal information. For example, a two-stage method may describe frames independently (“a person holds a cup” → “a cup is on the table”) while missing the action-level continuity (“the person puts down the cup”). This results in temporally fragmented supervision signals that limit the model’s ability to reason across time.
>
>   We will include illustrative examples comparing the single-stage and multi-stage synthesis approaches in the revised manuscript to better highlight their differences and clarify our design choice.
>
> - **On long-video generalization**
>
>   Thank you for the insightful comment. We would like to clarify that TempFlex-VL is not specifically designed for hour-scale or extremely long video understanding. Existing long-video MLLM models such as Video-XL, Video-XL-2, and ReTaKe typically adopt strategies like sparse frame sampling, keyframe selection, or visual token aggregation to efficiently handle extended temporal contexts.
>
>   In contrast, TempFlex-VL focuses on efficient visual encoding and enhanced spatio-temporal feature modeling, without using frame selection or compression strategies. Thanks to its efficient encoding design, TempFlex-VL also demonstrates a certain level of generalization to longer videos. We report its performance on the LongVideo benchmark in Table 3, where, under a uniform frame sampling strategy with 128 frames, TempFlex-Qwen3-4B achieves an accuracy of 60.4%, outperforming Video-XL-7B (50.7%).
>
>   Please note that the architecture of TempFlex is fully compatible with long-video MLLM's techniques such as frame selection and token compression. We will include a discussion of these potential integrations in the revised manuscript and consider them as promising directions for future work.
>
> **W2. Elaborate the comparison of inference latency**
> Thank you for your valuable comment. We agree that inference latency is an important factor to evaluate. Accordingly, we have added a comparison of TempFlex-VL with InternVL2.5-4B and Qwen2.5-VL-3B in terms of inference latency.
>
> For a fair comparison, all models were evaluated on a single NVIDIA H100 GPU using the same set of video samples (batch size = 1, maximum of 10 frames per video). We report the average Time to First Token (TTFT), calculated as the total inference time divided by the number of samples. The results are summarized below:
>
> | **Model**           | **TTFT (s)** |
> | -------------------- | ------------ |
> | InternVL2.5-4B       | 0.326        |
> | Qwen2.5-VL-3B        | 0.349        |
> | TempFlex-Gemma3-4B    |  0.311    |
> | TempFlex-Qwen3-4B    | **0.299**    |
>
> These results indicate that, benefiting from our efficient visual encoding strategy, TempFlex-VL achieves lower inference latency compared to both baselines. We will include a detailed discussion of these latency results in the revised manuscript.
>
>
> **W3. Improve presentation quality**
> Thank you for these detailed editorial suggestions and we will address them as follows:
> - Shorten the introduction by consolidating references and splitting the opening sentence for readability.
> - Use “MLLM” consistently throughout the paper to unify terminology.
> - Define abbreviations (e.g., TFF) only once upon first appearance.
> - Move Figure 4 (case visualization) before the conclusion section for smoother narrative flow.
>
>
> **Requested Changes:**
>
> Thank you for your valuable suggestion. We have addressed accordingly—please refer to our response in the Weakness part for details.
>
>
> We hope we have addressed your concerns satisfactorily. Please let us know if you have any new concerns or additional questions that we can assist with.

---

### Decision · Action_Editor_pmMM · 2025-11-09

**Recommendation:** Accept as is

**Audience:**

Yes

**Audience Explanation:**

Yes, the findings of this paper will be of interest to TMLR audience, because it addresses an important problem of video modeling efficiency, and demonstrates reasonable empirical gains. All reviewers agree this will be of interest to TMRL readers.

**Claims And Evidence:**

Yes

**Claims Explanation:**

Yes, the claims made in the submission were supported by clear experimental evidence including both main results on established image / video benchmarks and ablations of each proposed components. All reviewers agreed with this too.